# A Multi-Grained Group Symmetric Framework for Learning Protein-Ligand Binding Dynamics

## Abstract

In drug discovery, molecular dynamics (MD) simulation for protein-ligand binding provides a powerful tool for predicting binding affinities, estimating transport properties, and exploring pocket sites. There has been a long history of improving the efficiency of MD simulations through better numerical methods and, more recently, by augmenting them with machine learning (ML) methods. Yet, challenges remain, such as accurate modeling of extended-timescale simulations. To address this issue, we propose NeuralMD, the first ML surrogate that can facilitate numerical MD and provide accurate simulations of protein-ligand binding dynamics. We propose a principled approach that incorporates a novel physics-informed multi-grained group symmetric framework. Specifically, we propose (1) a BindingNet model that satisfies group symmetry using vector frames and captures the multi-level protein-ligand interactions, and (2) an augmented neural ordinary differential equation solver that learns the trajectory under Newtonian mechanics. For the experiment, we design ten single-trajectory and three multi-trajectory binding simulation tasks. We show the efficiency and effectiveness of NeuralMD, with a $2000\times$ speedup over standard numerical MD simulation and outperforming all other ML approaches by up to ~80% under the stability metric. We further qualitatively show that NeuralMD reaches more stable binding predictions.

## 1 Introduction

The simulation of protein-ligand binding dynamics is one of the fundamental tasks in drug discovery (Kairys et al., 2019; Yang et al., 2020; Volkov et al., 2022). Such simulations of binding dynamics are a key component of the drug discovery pipeline to select, refine, and tailor the chemical structures of potential drugs to enhance their efficacy and specificity. To simulate the protein-ligand binding dynamics, *numerical molecular dynamics (MD)* methods have been extensively developed. However, the numerical MD methods are computationally expensive due to the expensive force calculations on individual atoms in a large protein-ligand system.

To alleviate this issue, *machine learning (ML)* surrogates have been proposed to either augment or replace numerical MD methods to estimate the MD trajectories. However, all prior ML approaches for MD are limited to single-system dynamics (*e.g.*, small molecules or proteins) and not protein-ligand binding dynamics. A primary reason is the lack of large-scale datasets. The first large-scale dataset with binding dynamics was released in May 2023 (Siebenmorgen et al., 2023), and to our knowledge, we are now the first to explore it in this paper. Further, prior ML-based MD approaches limit to studying the MD dynamics on a small time interval (1e-15 seconds), while simulation on a longer time interval (*e.g.*, 1e-9 seconds) is needed for specific tasks, such as detecting the transient and cryptic states (Vajda et al., 2018) in binding dynamics. However, such longer-time MD simulations are challenging due to the catastrophic buildup of errors over longer rollouts (Fu et al., 2022a).

Another critical aspect that needs to be integrated into ML-based modeling is the group symmetry present in protein-ligand geometry. Specifically, the geometric function should be equivariant to rotation and translation (*i.e.*, SE(3)-equivariance). The principled approach to satisfy equivariance is to use vector frames, which have been previously explored for single molecules (Jumper et al.,

---

*Equal contribution

†Equal advising. Correspondence to *hongyu.guo@uottawa.ca, jchayes@berkeley.edu.*

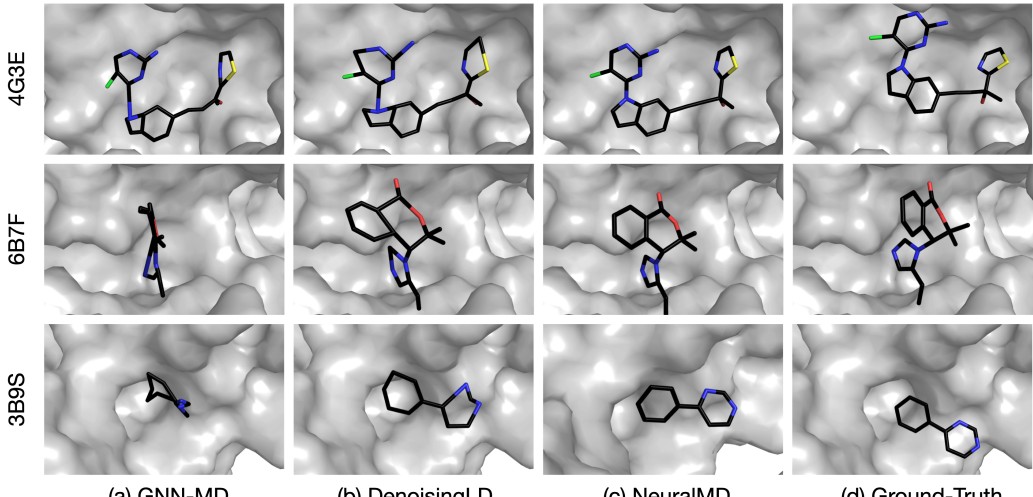

Figure 1: Visualization of last-snapshot binding predictions on three PDB complexes. NeuralMD stays more stable than DenoisingLD, exhibiting a lower degree of torsion with the natural conformations. Other methods collapse heavily, including GNN-MD and VerletMD, where atoms extend beyond the frame for the latter.

2021), but not yet for the protein-ligand binding complexes. The vector frame basis achieves SE(3)-equivariance by projecting vectors (*e.g.*, positions and accelerations) into the vector frame basis, and such a projection can maintain the equivariant property with efficient calculations (Liu et al., 2023a).

**Our Approach: NeuralMD.** We propose NeuralMD, a multi-grained physics-informed approach designed to handle extended-timestep MD simulations for protein-ligand binding dynamics. Our multi-grained method explicitly decomposes the complexes into three granularities: the atoms in ligands, the backbone structures in proteins, and the residue-atom pairs in the complex, to obtain a scalable approach for modeling a large system. We achieve **group symmetry** in BindingNet through the incorporation of vector frames, and include three levels of vector frame bases for multi-grained modeling, from the atom and backbone level to the residue level for binding interactions.

Further, our ML approach NeuralMD preserves the **Newtonian mechanics**. In MD, the movement of atoms is determined by Newton's second law, $F = m \cdot a$, where $F$ is the force, $m$ is the mass, and $a$ is the acceleration of each atom. By integrating acceleration and velocity w.r.t. time, we can obtain the velocities and positions, respectively. Thus in NeuralMD, we formulate the trajectory simulation as a second-order ordinary differential equation (ODE) problem, and it is solved using neural ODE.

**Experiments.** To verify the effectiveness and efficiency of NeuralMD, we design ten single-trajectory and three multi-trajectory binding simulation tasks. For evaluation, we adopt the recovery and stability metrics (Fu et al., 2022a). NeuralMD achieves 2000× speedup compared to the numerical methods. We observe that NeuralMD outperforms all other ML methods (Zhang et al., 2018; Musaelian et al., 2023b; Fu et al., 2022b; Wu & Li, 2023; Arts et al., 2023) on 12 tasks using recovery metric, and NeuralMD is consistently better by a large gap using the stability metric (up to ~80%). Qualitatively, we illustrate that NeuralMD realizes more stable binding dynamics predictions in three case studies. They are three protein-ligand binding complexes from Protein Data Bank (PDB), as shown in Figure 1.

**Related Work.** We briefly review the most related work here and include a more comprehensive discussion in Appendix A. **Geometric modelings** can be split into three categories (Liu et al., 2023a; Zhang et al., 2023): invariant models that utilize invariant features (distances and angles) (Schütt et al., 2018; Klicpera et al., 2020), equivariant models with spherical frames (Smidt et al., 2018; Musaelian et al., 2022), and equivariant models with vector frames (Jumper et al., 2021). This also includes the works studying the protein-ligand binding (Stepniewska-Dziubinska et al., 2018; Jiménez et al., 2018; Jones et al., 2021; Yang et al., 2023; Corso et al., 2023), but they are focusing on the equilibrium state, not the dynamics. On the other hand, existing ML works have studied **molecular simulation** (Zhang et al., 2018; Doerr et al., 2020; Musaelian et al., 2023b), and they can be categorized into two groups, each with its respective limitations. (1) The first direction focuses on predicting the energy (or force) (Zhang et al., 2018; Doerr et al., 2020; Musaelian et al., 2023b), which is then fed into the numerical integration algorithm for trajectory simulations. (2) Another research line directly predicts the atomic positions along the trajectories in an auto-regressive

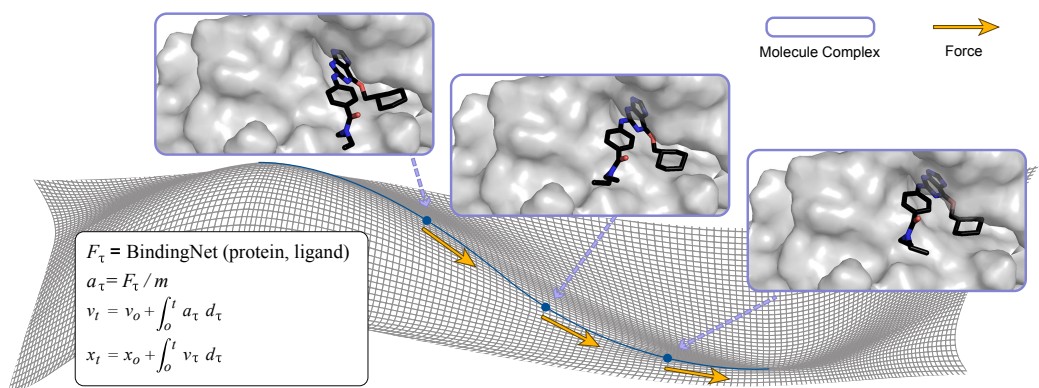

Figure 2: Illustrations of NeuralMD for protein-ligand binding dynamics. The landscape depicts the energy level, and the binding dynamic leads to an equilibrium state with lower energy.For NeuralMD, the force prediction follows the SE(3)-equivariance, and the dynamics prediction follows the Newtonian mechanics.

(AR) manner (Wu & Li, 2023; Arts et al., 2023; Fu et al., 2022b). Our work proposes a geometric ML framework as an MD simulation tool to learn the protein-ligand binding dynamics. It is also important to acknowledge the efforts of the whole community in constructing the datasets and benchmarks, including MISATO (Siebenmorgen et al., 2023) and Geom3D (Liu et al., 2023a).

## 2 PRELIMINIARIES

**Ligands.** In this work, we consider binding complexes involving small molecules as ligands. Small molecules can be treated as sets of atoms in the 3D Euclidean space, $\{f^{(l)}, x^{(l)}\}$, where $f^{(l)}$ and $x^{(l)}$ represent the atomic numbers and 3D Euclidean coordinates for atoms in each ligand, respectively.

**Proteins.** Proteins are macromolecules, which are essentially chains of amino acids or residues. There are 20 natural amino acids, and each amino acid is a small molecule. Noticeably, amino acids are made up of three components: a basic amino group (-NH$_2$), an acidic carboxyl group (-COOH), and an organic R group (or side chain) that is unique to each amino acid. Additionally, the carbon that connects all three groups is called C$_\alpha$. Due to the large number of atoms in proteins, this work proposes a multi-grained method for modeling the protein-ligand complexes. In this regard, the **backbone-level** data structure for each protein is $\{f^{(p)}, \{x_N^{(p)}, x_{C_\alpha}^{(p)}, x_C^{(p)}\}\}$, for the residue type and the coordinates of $N - C_\alpha - C$ in each residue, respectively. (We may omit the superscript in the coordinates of backbone atoms, as these backbone structures are unique to protein residues.) In addition to the backbone level, for a coarser-grained data structure on protein-ligand complex, we further consider **residue-level** modeling for binding interactions, $\{f^{(p)}, x^{(p)}\}$, where the coordinate of C$_\alpha$ is taken as the residue-level coordinate, *i.e.*, $x^{(p)} \triangleq x_{C_\alpha}^{(p)}$.

**Molecular Dynamics Simulations.** Generally, molecular dynamics (MD) describes how each atom in a molecular system moves over time, following Newton's second law of motion:

$$F = m \cdot a = m \cdot \frac{d^2 x}{dt^2}, \tag{1}$$

where $F$ is the force, $m$ is the mass, $a$ is the acceleration, $x$ is the position, and $t$ is the time. Then, an MD simulation will take Newtonian dynamics, an ordinary differential equation (ODE), to get the trajectories, where such a molecular system can be a small molecule, a protein, a polymer, or a protein-ligand binding complex. The **numerical methods for MD** can be classified into classical MD and *ab-initio* MD, where the difference lies in how the force on each atom is calculated: *ab-initio* MD calculates the forces using a quantum-mechanics-based method, such as density functional theory (DFT), while classical MD uses force-field-based approaches to predict the atomic forces. More recently, **machine learning (ML) methods for MD** have opened a new perspective by utilizing the group symmetric tools for geometric representation and the automatic differential tools for dynamics learning (please check Appendix A for a more detailed discussion).

In MD simulations, the systems are considered to be either in a vacuum or with explicit modeling of solvent or air molecules. The former is impractical in a real-world system, especially when the jostling

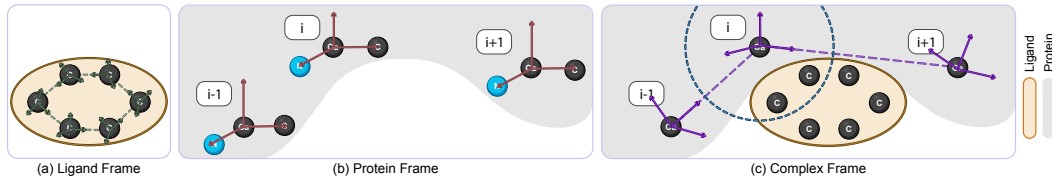

Figure 3: Three granularities of vector frame basis in BindingNet: (a) atom-level basis for ligands, (b) backbone-level basis for proteins, and (c) residue-level basis for the protein-ligand complex.

of solvent or air molecules causes friction, and the occasional high-velocity collision can perturb the system, leading to prohibitively high computational costs. Thus, as a surrogate solution, researchers have considered adopting Brownian dynamics or Langevin dynamics for molecular simulation (Wu & Li, 2023): they can reduce the computational cost while keeping track of the stochasticity of solvent or air molecules. In the ML-based MD simulations, adopting Newtonian dynamics or Langevin dynamics can be treated as a selection of different inductive biases when building the framework. In this work, we follow Newtonian dynamics since the experiment dataset (MISATO (Siebenmorgen et al., 2023)) uses classical MD, explicitly including all the solvent molecules.

**Problem Setting: Protein-Ligand Binding Dynamics Simulation.** In this work, we are interested in the protein-ligand binding dynamics in the semi-flexible setting (Salmaso & Moro, 2018), *i.e.*, proteins with rigid structures and ligands with flexible movements (Siebenmorgen et al., 2023; Corso et al., 2023). Thus, the problem is formulated as follows: suppose we have a fixed protein structure $\{f^{(p)}, \{x_N^{(p)}, x_{C_\alpha}^{(p)}, x_C^{(p)}\}\}$ and a ligand with its initial structure and velocity, $\{f^{(l)}, x_0^{(l)}, v_0^{(l)}\}$. We want to predict the trajectories of ligands following the Newtonian dynamics, *i.e.*, the movement of $\{x_t^{(l)}, ...\}$ over time. We also want to clarify two critical points about this problem setting: (1) Our task is trajectory prediction, *i.e.*, positions as labels, and no explicit energy and force labels are considered. ML methods for energy prediction followed with numerical ODE solver may require smaller time intervals (*e.g.*, 1e-15 seconds), while trajectory prediction, which directly predicts the positions, is agnostic to the magnitude of time intervals. This is appealing for datasets like MISATO with larger time intervals (*e.g.*, 1e-9 seconds), as will be discussed in Section 4. (2) Each trajectory is composed of a series of geometries of molecules, and such geometries are called **snapshots**. We avoid using *frames* since we will introduce the *vector frame* in modeling the binding complex in Section 3.

## 3 METHOD: BINDINGNET AND NEURALMD

In this section, we will introduce our framework, NeuralMD, for simulating the protein-ligand binding dynamics. It has two main phases: (1) A multi-grained SE(3)-equivariant geometric model, BindingNet. It models the protein-ligand complex from three granularities, employing three levels of vector frame basis: atom level for ligands, backbone level for proteins, and residue level for protein-ligand complexes. (2) A second-order ordinary differential equation (ODE) solver to learn the Newtonian mechanics. We propose a variant of neural ODE with two attributes (Chen et al., 2018): the ODE augmentation for the efficient second-order solver and the joint ODE for lower memory costs. This section outlines the structure: the three levels of vector frames in Section 3.1, the architecture of BindingNet in Section 3.2, the design of NeuralMD in Section 3.3, and implementation details in Section 3.4.

### 3.1 MULTI-GRAINED VECTOR FRAMES

Proteins are essentially macromolecules composed of thousands of residues (amino acids), where each residue is a small molecule. Thus, it is infeasible to model all the atoms in proteins due to the large volume of the system, and such an issue also holds for the protein-ligand complex. To address this issue, we propose BindingNet, a multi-grained SE(3)-equivariant model, to capture the interactions between a ligand and a protein. The vector frame basis ensures SE(3)-equivariance, and the multi-granularity is achieved by considering frames at three levels.

**Vector Frame Basis for SE(3)-Equivariance.** Recall that the geometric representation of the whole molecular system needs to follow the physical properties of the equivariance w.r.t. rotation and translation. Such a group symmetric property is called SE(3)-equivariance. We also want to point out that the reflection or chirality property is equivariant for properties like energy, yet it is not for

the ligand modeling with rigid protein structures (*i.e.*, antisymmetric to the reflection). The vector frame basis can handle this naturally, and we leave a more detailed discussion in Appendix C, along with the proof on group symmetry of vector frame basis. In the following, we introduce three levels of vector frames for multi-grained modeling.

**Atom-Level Vector Frame for Ligands.** For small molecule ligands, we first extract atom pairs $(i, j)$ within the distance cutoff $c$, and the vector frame basis is constructed using the Gram-Schmidt as:

$$\mathcal{F}_{\text{ligand}} = \Big( \frac{\boldsymbol{x}_i^{(l)} - \boldsymbol{x}_j^{(l)}}{\left\| \boldsymbol{x}_i^{(l)} - \boldsymbol{x}_j^{(l)} \right\|}, \frac{\boldsymbol{x}_i^{(l)} \times \boldsymbol{x}_j^{(l)}}{\left\| \boldsymbol{x}_i^{(l)} \times \boldsymbol{x}_j^{(l)} \right\|}, \frac{\boldsymbol{x}_i^{(l)} - \boldsymbol{x}_j^{(l)}}{\left\| \boldsymbol{x}_i^{(l)} - \boldsymbol{x}_j^{(l)} \right\|} \times \frac{\boldsymbol{x}_i^{(l)} \times \boldsymbol{x}_j^{(l)}}{\left\| \boldsymbol{x}_i^{(l)} \times \boldsymbol{x}_j^{(l)} \right\|} \Big), \tag{2}$$

where $\times$ is the cross product. Note that both $\boldsymbol{x}_i^{(l)}$ and $\boldsymbol{x}_j^{(l)}$ are for geometries at time $t$ - henceforth, we omit the subscript $t$ for brevity. Such an atom-level vector frame allows us to do SE(3)-equivariant message passing to get the atom-level representation.

**Backbone-Level Vector Frame for Proteins.** Proteins can be treated as chains of residues, where each residue possesses a backbone structure. The backbone structure comprises an amino group, a carboxyl group, and an alpha carbon, delegated as $N - C_\alpha - C$. Such a structure serves as a natural way to build the vector frame. For each residue in the protein, the coordinates are $\boldsymbol{x}_N$, $\boldsymbol{x}_{C_\alpha}$, and $\boldsymbol{x}_C$, then the backbone-level vector frame for this residue is:

$$\mathcal{F}_{\text{protein}} = \Big( \frac{\boldsymbol{x}_N - \boldsymbol{x}_{C_\alpha}}{\| \boldsymbol{x}_N - \boldsymbol{x}_{C_\alpha} \|}, \frac{\boldsymbol{x}_{C_\alpha} - \boldsymbol{x}_C}{\| \boldsymbol{x}_{C_\alpha} - \boldsymbol{x}_C \|}, \frac{\boldsymbol{x}_N - \boldsymbol{x}_{C_\alpha}}{\| \boldsymbol{x}_N - \boldsymbol{x}_{C_\alpha} \|} \times \frac{\boldsymbol{x}_{C_\alpha} - \boldsymbol{x}_C}{\| \boldsymbol{x}_{C_\alpha} - \boldsymbol{x}_C \|} \Big). \tag{3}$$

This is built for each residue, providing a residue-level representation.

**Residue-Level Vector Frame for Protein-Ligand Complexes.** It is essential to model the protein-ligand interaction to better capture the binding dynamics. We achieve this by introducing the residue-level vector frame. More concretely, proteins are sequences of residues, marked as $\{(f_0^{(p)}, \boldsymbol{x}_0^{(p)}), ..., (f_i^{(p)}, \boldsymbol{x}_i^{(p)}), (f_{i+1}^{(p)}, \boldsymbol{x}_{i+1}^{(p)}, ...\}$. Here, we use a cutoff threshold $c$ to determine the interactions between ligands and proteins, and the interactive regions on proteins are called pockets. We construct the following vector frame for residues in the pockets sequentially:

$$\mathcal{F}_{\text{complex}} = \Big( \frac{\boldsymbol{x}_i^{(p)} - \boldsymbol{x}_{i+1}^{(p)}}{\left\| \boldsymbol{x}_i^{(p)} - \boldsymbol{x}_{i+1}^{(p)} \right\|}, \frac{\boldsymbol{x}_i^{(p)} \times \boldsymbol{x}_{i+1}^{(p)}}{\left\| \boldsymbol{x}_i^{(p)} \times \boldsymbol{x}_{i+1}^{(p)} \right\|}, \frac{\boldsymbol{x}_i^{(p)} - \boldsymbol{x}_{i+1}^{(p)}}{\left\| \boldsymbol{x}_i^{(p)} - \boldsymbol{x}_{i+1}^{(p)} \right\|} \times \frac{\boldsymbol{x}_i^{(p)} \times \boldsymbol{x}_{i+1}^{(p)}}{\left\| \boldsymbol{x}_i^{(p)} \times \boldsymbol{x}_{i+1}^{(p)} \right\|} \Big). \tag{4}$$

Through this complex-level vector frame, the message passing enables the exchange of information between atoms from ligands and residues from the pockets. The illustration of the above three levels of vector frames can be found in Figure 3. Once we build up such three vector frames, we then conduct a *scalarization* operation (Hsu, 2002), which transforms the equivariant variables (*e.g.*, coordinates) to invariant variables by projecting them to the three vector bases in the vector frame.

### 3.2 Multi-Grained SE(3)-Equivariant Binding Force Modeling: BindingNet

In this section, we introduce BindingNet, a multi-grained SE(3)-equivariant geometric model for protein-ligand binding. The input of BindingNet is the geometry of the rigid protein and the ligand at time $t$, while the output is the force on each atom in the ligand.

**Atom-Level Ligand Modeling.** We first generate the atom embedding using one-hot encoding and then aggregate each atom's embedding, $\boldsymbol{z}^{(l)}$, by aggregating all its neighbor's embedding within the cutoff distance $c$. Then, we obtain the atom's equivariant representation by aggregating its neighborhood's messages as $(\boldsymbol{x}_i^{(l)} - \boldsymbol{x}_j^{(l)}) \cdot \boldsymbol{z}_i^{(l)}$. A subsequent scalarization is carried out based on the atom-level vector frame as $\boldsymbol{h}_{ij}^{(l)} = (\boldsymbol{h}_i^{(l)} \oplus \boldsymbol{h}_j^{(l)}) \cdot \mathcal{F}_{\text{ligand}}$, where $\oplus$ is the concatenation. Finally, it is passed through several equivariant message-passing layers (MPNN) defined as:

$$\text{vec}_i^{(l)} = \text{vec}_i^{(l)} + \text{Agg}_j \big( \text{vec}_i^{(l)} \cdot \text{MLP}(\boldsymbol{h}_{ij}) + (\boldsymbol{x}_i^{(l)} - \boldsymbol{x}_j^{(p)}) \cdot \text{MLP}(\boldsymbol{h}_{ij}) \big), \tag{5}$$

where $\text{MLP}(\cdot)$ and $\text{Agg}(\cdot)$ are the multi-layer perceptron and mean aggregation functions, respectively. $\text{vec} \in \mathbb{R}^3$ is a vector assigned to each atom and is initialized as 0. The outputs are atom representation and vector ($\boldsymbol{h}^{(l)}$ and $\text{vec}^{(l)}$), and they are passed to the complex module.

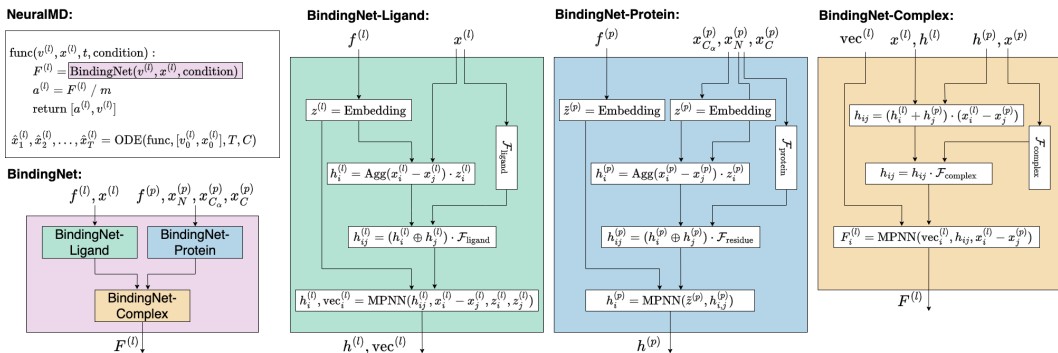

Figure 4: Brief pipeline of NeuralMD. In the three key modules of BindingNet, there are three vertical boxes, corresponding to three granularities of vector frames, as in Equations (2) to (4). More details are in Appendix E.

**Backbone-Level Protein Modeling.** For the coarse-grained modeling of proteins, we consider three backbone atoms in each residue. We first obtain the atom embedding on three atom types, and then we obtain each atom's representation $z^{(p)}$ by aggregating its neighbor's representation. Then, we obtain an equivariant atom representation by aggregating the edge information, $(x_i^{(p)} - x_j^{(p)}) \cdot z_i^{(p)}$, within cutoff distance $c$. Following which is the scalarization on the residue frame $h_{ij}^{(p)} = (h_i^{(p)} \oplus h_j^{(p)}) \cdot \mathcal{F}_{\text{protein}}$. Recall that we also have the residue type, and with a type embedding $\tilde{z}^{(p)}$, we can obtain the final residue-level representation using an MPNN layer as $h^{(p)} = \tilde{z}^{(p)} + (h_{N,C_\alpha}^{(p)} + h_{C_\alpha,C}^{(p)})/2$.

**Residue-Level Complex Modeling.** Once we obtain the atom-level representation and vector $(h^{(l)}, \text{vec}^{(l)})$ from ligands, and backbone-level representation $(h^{(p)})$ from proteins, the next step is to learn the protein-ligand interaction. We first extract the residue-atom pair $(i, j)$ with a cutoff $c$, based on which we obtain an equivariant interaction edge representation $h_{ij} = (h_i^{(l)} + h_j^{(p)}) \cdot (x_i^{(l)} - x_j^{(p)})$. After scalarization, we can obtain invariant interaction edge representation $h_{ij} = h_{ij} \cdot \mathcal{F}_{\text{complex}}$. Finally, we adopt an equivariant MPNN layer to get the atom-level force as:

$$\text{vec}_{ij}^{(pl)} = \text{vec}_i^{(l)} \cdot \text{MLP}(h_{ij}) + (x_i^{(l)} - x_j^{(p)}) \cdot \text{MLP}(h_{ij}), \quad F_i^{(l)} = \text{vec}_i^{(l)} + \text{Agg}_{j \in \mathcal{N}(i)} \text{vec}_{ij}^{(pl)}. \quad (6)$$

In the last equation, the ultimate force predictions can be viewed as two parts: the internal force from the molecule $\text{vec}_i^{(l)}$ and the external force from the protein-ligand interaction $\text{vec}_{ij}^{(pl)}$. More complete descriptions of the three modules can be found in Appendix E.

## 3.3 BINDING MOLECULAR DYNAMICS MODELING: NEURALMD

As clarified in Section 2, molecular dynamics follows Newtonian dynamics, and we solve it as an ordinary differential equation (ODE) problem. The BindingNet introduced in Sections 3.1 and 3.2 takes in the molecular system geometry $(x_t^{(l)}, x^{(p)})$ at arbitrary time $t$, and outputs the forces. Then in this section, we describe how we use the neural ODE solver to predict the coordinates at future snapshots.

We want to highlight that one research line is predicting the force field directly (Zhang et al., 2018; Doerr et al., 2020; Musaelian et al., 2023b), which will be fed into the numerical integration algorithms for trajectory simulation. For accuracy, such a simulation must be at the femtosecond level (1e-15). However, as shown in recent works (Unke et al., 2021; Stocker et al., 2022; Fu et al., 2022a), minor errors in the force field can lead to catastrophic failure for long-time simulations. For instance, there can be pathological behavior, such as extreme force predictions or bond breaking, within the low end of the distribution. Our experiments have yielded similar observations, as will be shown in Section 4. In this paper, however, we overcome this issue by directly learning the extended-timescale MD trajectories (nanosecond level, 1e-9).

On the other hand, recent explorations on automatic differentiation have opened new perspectives for solving the ordinary/partial differential equations (ODE/PDEs) (Chen et al., 2018; Raissi et al., 2019; Lu et al., 2019; 2021; Li et al., 2020; Kovachki et al., 2021). Newtonian mechanics in MD is essentially an ODE, and for protein-ligand complexes with a large volume of atoms, we consider using adjoint neural ODE (NODE) to save memory costs (Chen et al., 2018), allowing small step sizes

in the ODE integration. Roughly speaking, the adjoint method calculates the variational derivative of the trajectory w.r.t. our objective function directly. Thus, it avoids calling the auto-differentiation for all intermediate time intervals. We further want to highlight that we are working on a second-order ODE in Newtonian dynamics. The key module of NODE is the ODE function, which returns the first-order derivative. Then, it will be integrated using algorithms like Euler. To learn the MD trajectory following second-order ODE, we propose the following formulation of the second-order ODE within one integration call:

$$\begin{bmatrix} d\boldsymbol{x}/dt \\ d\boldsymbol{v}/dt \end{bmatrix} = \begin{bmatrix} \boldsymbol{v} \\ F/m \end{bmatrix}, \tag{7}$$

where $F$ is the output forces from BindingNet. This means we augment ODE derivative space by concurrently calculating the accelerations and velocities, allowing simultaneous integration of velocities and positions. Ultimately, following Newtonian mechanics, the coordinates at time $t$ are integrated as:

$$\begin{aligned} F_\tau^{(l)} &= \text{BindingNet}(f^{(l)}, \boldsymbol{x}_\tau^{(l)}, f^{(p)}, \boldsymbol{x}_N^{(p)}, \boldsymbol{x}_{C_\alpha}^{(p)}, \boldsymbol{x}_C^{(p)}), & \boldsymbol{a}_\tau^{(l)} &= \frac{F_\tau^{(l)}}{m}, \\ \hat{\boldsymbol{v}}_t^{(l)} &= \boldsymbol{v}_0^{(l)} + \int_0^t \boldsymbol{a}_\tau^{(l)} d\tau, & \hat{\boldsymbol{x}}_t^{(l)} &= \boldsymbol{x}_0^{(l)} + \int_0^t \hat{\boldsymbol{v}}_\tau^{(l)} d\tau. \end{aligned} \tag{8}$$

The objective is the mean absolute error between the predicted coordinates and ground-truth coordinates: $\mathcal{L} = \mathbb{E}_t\big[|\hat{\boldsymbol{x}}_t^{(l)} - \boldsymbol{x}_t^{(l)}|\big]$. An illustration of NeuralMD pipeline is in Figure 4.

**From Newtonian dynamics to Langevin dynamics.** It is worth noting that the protein-ligand binding dynamics in the MISATO dataset considers the solvent molecules. However, the released dataset does not include the trajectory information of solvent molecules. To compensate such missing information, we leverage Langevin dynamics. Specifically, Langevin dynamics is an extension of the standard Newtonian dynamics with the addition of damping and random noise terms: $\hat{F}_\tau^{(l)} =$ BindingNet($f^{(l)}, \boldsymbol{x}_\tau^{(l)}, f^{(p)}, \boldsymbol{x}_N^{(p)}, \boldsymbol{x}_{C_\alpha}^{(p)}, \boldsymbol{x}_C^{(p)}) - \gamma m \boldsymbol{v} + \sqrt{2m\gamma k_B T} R(t)$, where $\gamma$ is the damping constant or collision frequency, $T$ is the temperature, $k_B$ is the Boltzmann's constant, and $R(t)$ is a delta-correlated stationary Gaussian process with zero-mean. This modified $\hat{F}_\tau^{(l)}$ is then plugged into Equation (8) to train and sample the trajectories, thus compensating the missing solvent effect.

### 3.4 Implementation Details and Summary

In this section, we would like to include extra details of NeuralMD. As in Equations (7) and (8), both the coordinates and velocities are required inputs for NeuralMD. Unfortunately, certain datasets may not cover such information. To handle this issue, we propose a surrogate velocity, which is a summation of a predicted velocity by an extra equivariant model and a coordinate momentum, *i.e.*, $\boldsymbol{v}_t^{(l)} = \text{BindingNet-Ligand}(f^{(l)}, \boldsymbol{x}_t^{(l)}) + (\boldsymbol{x}_{t+1}^{(l)} - \boldsymbol{x}_t^{(l)})$, where the BindingNet-Ligand is the ligand model described in Section 3.2. This is not displayed in Figure 4 for brevity, yet we provide all the critical details for reproducibility. The algorithm of our method is presented in Algorithm 1.

---

**Algorithm 1** Training of NeuralMD

1: **Input:** Initial position $\boldsymbol{x}_0^{(l)}$ and initial velocity $\boldsymbol{v}_0^{(l)}$ for ligands, atomic features $f^{(l)}$ for ligands, residue types and coordinates $f^{(p)}, \boldsymbol{x}_N^{(p)}, \boldsymbol{x}_{C_\alpha}^{(p)}, \boldsymbol{x}_C^{(p)}$ for proteins, and time $T$.
2: **for** discretized time $t \in \{1, 2, ..., T-1\}$ **do**
3:     Centralize the coordinates of the ligand-protein complex for $\boldsymbol{x}_0^{(l)}$ and $\boldsymbol{x}^{(p)}$ by removing the mass center.
4:     Set T_list = $[t, t+1, ...]$ and condition $C = [f^{(l)}, f^{(p)}, \boldsymbol{x}_N^{(p)}, \boldsymbol{x}_{C_\alpha}^{(p)}, \boldsymbol{x}_C^{(p)}]$.
5:     Get predicted position $[\hat{\boldsymbol{x}}_{t+1}^{(l)}, \hat{\boldsymbol{x}}_{t+2}^{(l)}, ...] = \text{ODE}(\text{NeuralMD}, [\boldsymbol{x}_0^{(l)}, \boldsymbol{v}_0^{(l)}], \text{T\_list}, C)$.     // Equation (7)
6:     Calculate the position prediction loss $\mathcal{L} = \mathbb{E}_\tau\big[\text{MAE}(\hat{\boldsymbol{x}}_\tau^{(l)}, \boldsymbol{x}_\tau^{(l)})\big]$.
7:     Update model BindingNet using gradient descent.
8: **end for**

---

## 4 Experiments

### 4.1 Experiment Setting

**Datasets.** As described in Appendix A, one of the main bottlenecks of ML for binding dynamics simulation is the available datasets for learning molecular dynamics (MD). Recently, the community

has put more effort into gathering the datasets, and we consider MISATO in our work (Siebenmorgen et al., 2023). It is built on 16,972 experimental protein-ligand complexes extracted from the protein data bank (PDB) (Berman et al., 2000). Such data is obtained using X-ray crystallography, Nuclear Magnetic Resonance(NMR), or Cryo-Electron Microscopy (Cryo-EM), where systematic errors are unavoidable. This motivates the MISATO project, which utilizes semi-empirical quantum mechanics for structural curation and refinement, including regularization of the ligand geometry. For each protein-ligand complex, the trajectory comprises 100 snapshots in 8 nanoseconds under the fixed temperature and pressure. We want to highlight that MD trajectories allow the analysis of small-range structural fluctuations of the protein-ligand complex. Still, in some cases, large-scale rare events can be observed. In Appendix D, we list several basic statistics of MISATO, *e.g.*, the number of atoms in small molecule ligands and the number of residues in proteins.

**Baselines.** In this work, we mainly focus on machine learning methods for trajectory prediction, *i.e.*, no energy or force labels are considered. GNN-MD is to apply geometric graph neural networks (GNNs) to predict the trajectories in an auto-regressive manner (Siebenmorgen et al., 2023; Fu et al., 2022b). More concretely, GNN-MD takes as inputs the geometries at time $t$ and predicts the geometries at time $t + 1$. DenoisingLD (denoising diffusion for Langevin dynamics) (Arts et al., 2023; Wu & Li, 2023; Fu et al., 2022b) is a baseline method that models the trajectory prediction as denoising diffusion task (Song et al., 2020), and the inference for trajectory generation essentially becomes the Langevin dynamics. CG-MD learns a dynamic GNN and a score GNN (Fu et al., 2022b), which are essentially the hybrid of GNN-MD and DenoisingLD. Here, to make the comparison more explicit, we compare these two methods (GNN-MD and DenoisingLD) separately. Additionally, we consider VerletMD, an energy prediction research line (including DeePMD (Zhang et al., 2018), TorchMD (Doerr et al., 2020), and Allegro-LAMMPS (Musaelian et al., 2023b)), where the role of ML models is to predict the energy, and the MD trajectory is obtained by the velocity Verlet algorithm, a numerical integration method for Newtonian mechanics. We keep the same backbone model (BindingNet) for energy or force prediction for all the baselines.

**Experiments Settings.** We consider two experiment settings. The first type of experiment is the single-trajectory prediction, where both the training and test data are snapshots from the same trajectory, and they are divided temporally. The second type of experiment is the multi-trajectory prediction, where each data point is the sequence of all the snapshots from one trajectory, and the training and test data correspond to different sets of trajectories.

**Evaluation Metrics.** For MD simulation, the evaluation is a critical factor for evaluating trajectory prediction (Fu et al., 2022a). For both experiment settings, the trajectory recovery is the most straightforward evaluation metric. To evaluate this, we take both the mean absolute error (MAE) and mean squared error (MSE) between the predicted coordinates and ground-truth coordinates over all snapshots. Stability, as highlighted in (Fu et al., 2022a), is an important metric for evaluating the predicted MD trajectory. The intuition is that the prediction on long-time MD trajectory can enter a pathological state (*e.g.*, bond breaking), and stability is the measure to quantify such observation. It is defined as $\mathbb{P}_{i,j}\big|\|\boldsymbol{x}_i - \boldsymbol{x}_j\| - \boldsymbol{b}_{i,j}\big| > \Delta$, where $\boldsymbol{b}_{i,j}$ is the pair distance at the last snapshot (the most equilibrium state), and we take $\Delta = 0.5$ Å. Another metric considered is frames per second (FPS) (Fu et al., 2022a) on a single Nvidia-V100 GPU card, and it measures the MD efficiency.

**Ablation Study on Flexible Setting.** As introduced in Section 2, we have been focusing on the semi-flexible setting for binding dynamics so far. Yet, we also conduct ablation studies on the flexible setting for small-scale experiments, as a proof-of-concept. For more details, please check Appendix F.

## 4.2 MD Prediction: Generalization On One Single Trajectory

This type of task has been widely studied in the existing literature for other molecular systems. Specifically, both the training and test data are from the same trajectory of one single protein-ligand complex, and here we take the first 80 snapshots for training and the remaining 20 snapshots for test.

Results are in Table 1. The first observation is that the baseline VertletMD has a clear performance gap compared to the other methods. This verifies that using ML models to predict the energy (or force) at each snapshot, and then using a numerical integration algorithm can fail in the long-time simulations (Fu et al., 2022a). Additionally, we can observe that the recovery error of trajectory (MAE and MSE) occasionally fails to offer a discernible distinction among methods (*e.g.*, for protein-ligand complex 3EOV, 1KT1, and 4G3E), though NeuralMD is slightly better. However, the stability (%) can

Table 1: Results on ten single-trajectory binding dynamics prediction. Results with optimal training loss are reported. Four evaluation metrics are considered: MAE (Å, ↓), MSE (↓), and Stability (%, ↓).

| PDB ID | Metric | VerletMD | GNN-MD | DenoisingLD | NeuralMD ODE (Ours) | NeuralMD SDE (Ours) | PDB ID | Metric | VerletMD | GNN-MD | DenoisingLD | NeuralMD ODE (Ours) | NeuralMD SDE (Ours) |
|---|---|---|---|---|---|---|---|---|---|---|---|---|---|
| 5WIJ | MAE | 9.618 | 2.319 | 2.254 | 2.118 | **2.109** | 1XP6 | MAE | 13.444 | 2.303 | 1.915 | **1.778** | 1.822 |
|  | MSE | 6.401 | 1.553 | 1.502 | 1.410 | **1.408** |  | MSE | 9.559 | 1.505 | 1.282 | **1.182** | 1.216 |
|  | Stability | 79.334 | 45.369 | 18.054 | **12.654** | 13.340 |  | Stability | 86.393 | 43.019 | 28.417 | **19.256** | 22.734 |
| 4ZX0 | MAE | 21.033 | 2.255 | 1.998 | **1.862** | 1.874 | 4YUR | MAE | 15.674 | 7.030 | 6.872 | **6.807** | 6.826 |
|  | MSE | 14.109 | 1.520 | 1.347 | **1.260** | 1.271 |  | MSE | 10.451 | 4.662 | 4.520 | **4.508** | 4.526 |
|  | Stability | 76.878 | 41.332 | 23.267 | **18.189** | 18.845 |  | Stability | 81.309 | 50.238 | 32.423 | **23.250** | 25.008 |
| 3EOV | MAE | 25.403 | 3.383 | 3.505 | 3.287 | **3.282** | 4G3E | MAE | 5.181 | 2.672 | 2.577 | 2.548 | **2.478** |
|  | MSE | 17.628 | 2.332 | 2.436 | 2.297 | **2.294** |  | MSE | 3.475 | 1.743 | 1.677 | 1.655 | **1.615** |
|  | Stability | 91.129 | 57.363 | 51.590 | **44.775** | 44.800 |  | Stability | 65.377 | 16.365 | 7.188 | **2.113** | 2.318 |
| 4K6W | MAE | 14.682 | 3.674 | 3.555 | 3.503 | **3.429** | 6B7F | MAE | 31.375 | 4.129 | 3.952 | 3.717 | **3.657** |
|  | MSE | 9.887 | 2.394 | 2.324 | 2.289 | **2.234** |  | MSE | 21.920 | 2.759 | 2.676 | 2.503 | **2.469** |
|  | Stability | 87.147 | 57.852 | 39.580 | 38.562 | **38.476** |  | Stability | 87.550 | 54.900 | 16.050 | **3.625** | 22.750 |
| 1KTI | MAE | 18.067 | **6.534** | 6.657 | 6.548 | 6.537 | 3B9S | MAE | 19.347 | 2.701 | 2.464 | **2.351** | 2.374 |
|  | MSE | 12.582 | 4.093 | 4.159 | 4.087 | **4.085** |  | MSE | 11.672 | 1.802 | 1.588 | **1.527** | 1.542 |
|  | Stability | 77.315 | 4.691 | 7.377 | 0.525 | **0.463** |  | Stability | 41.667 | 43.889 | 8.819 | **0.000** | 0.000 |

Table 2: Efficiency comparison of FPS between VerletMD and NeuralMD on single-trajectory prediction.

| PDB ID | 5WIJ | 4ZX0 | 3EOV | 4K6W | 1KTI | 1XP6 | 4YUR | 4G3E | 6B7F | 3B9S | Average |
|---|---|---|---|---|---|---|---|---|---|---|---|
| VerletMD | 12.564 | 30.320 | 29.890 | 26.011 | 19.812 | 28.023 | 31.513 | 29.557 | 19.442 | 31.182 | 25.831 |
| NeuralMD (Ours) | 33.164 | 39.415 | 31.720 | 31.909 | 24.566 | 37.135 | 39.365 | 39.172 | 20.320 | 37.202 | 33.397 |

Table 3: Results on three multi-trajectory binding dynamics predictions. Results with optimal validation loss are reported. Four evaluation metrics are considered: MAE (Å, ↓), MSE (↓), and Stability (%, ↓).

| Dataset | MISATO-100 | | | MISATO-1000 | | | MISATO-All | | |
|---|---|---|---|---|---|---|---|---|---|
|  | MAE | MSE | Stability | MAE | MSE | Stability | MAE | MSE | Stability |
| VerletMD | 90.326 | 56.913 | 86.642 | 80.187 | 53.110 | 86.702 | 105.979 | 69.987 | 90.665 |
| GNN-MD | 7.176 | 4.726 | 35.431 | 7.787 | 5.118 | 33.926 | 8.260 | 5.456 | 32.638 |
| DenoisingLD | 7.112 | 4.684 | 29.956 | 7.746 | 5.090 | 18.898 | 15.878 | 10.544 | 89.586 |
| NeuralMD-ODE (Ours) | **6.852** | **4.503** | **19.173** | **7.653** | **5.028** | **15.572** | **8.147** | **5.386** | **17.468** |
| NeuralMD-SDE (Ours) | 6.869 | 4.514 | 19.561 | 7.665 | 5.037 | 16.501 | 8.165 | 5.398 | 19.012 |

be a distinctive factor in method comparisons, where we observe NeuralMD outperform on all 10 tasks up to ~80%. We further qualitatively show the predicted trajectories of three case studies using three methods and ground truth in Figure 1. It is shown that the GNN-MD collapses occasionally, while DenoisingLD stays comparatively structured. Meanwhile, NeuralMD is the most stable in all cases.

One main benefit of using NeuralMD for binding simulation is its efficiency. To show this, we list the computational time in Table 2. We further approximate the wall time of the numerical method for MD simulation (PDB 5WIJ). Concretely, we can get an estimated speed of 1 nanosecond of dynamics every 0.28 hours. This is running the simulation with GROMACS (Van Der Spoel et al., 2005) on 1 GPU with 16 CPU cores and a moderately sized water box at the all-atom level (with 2 femtosecond timesteps). This equivalently shows that NeuralMD is ~2000× faster than numerical methods.

### 4.3 MD Prediction: Generalization Among Multiple Trajectories

A more challenging task is to test the generalization ability of NeuralMD among different trajectories. The MISATO dataset includes 13,765 protein-ligand complexes, and we first create two small datasets by randomly sampling 100 and 1k complexes, respectively. Then, we take 80%-10%-10% for training, validation, and testing. We also consider the whole MISATO dataset, where the data split has already been provided. After removing the peptide ligands, we have 13,066, 1,357, and 1,357 complexes for training, validation, and testing, respectively.

The quantitative results are in Table 3. First, we can observe that VerletMD has worse performance on all three datasets, and the performance gap with other methods is even larger compared to the single-trajectory prediction. The other two baselines, GNN-MD and DenoisingLD, show similar performance, while NeuralMD outperforms in all datasets. Notice that stability (%) remains more distinguishable than the two trajectory recovery metrics (MAE and MSE).

## 5 Conclusion and Outlook

To sum up, we devise NeuralMD, an ML framework that incorporates a novel multi-grained group symmetric network architecture and second-order ODE Newtonian dynamics, enabling accurate predictions of protein-ligand binding dynamics in a larger time interval. Not only is such a time

interval critical for understanding the dynamic nature of the ligand-protein complex, but our work marks the first approach to learning to predict binding dynamics. We quantitatively and qualitatively verify that NeuralMD achieves superior performance on 13 binding prediction tasks.

One potential limitation of our work is the dataset. Currently, we are using the MISATO dataset, a binding simulation dataset with a large timescale. However, NeuralMD is agnostic to the time interval, and it can also be applied to binding dynamics datasets with time interval as a femtosecond. This is beyond the scope of this work and may require the effort of the whole community for the dataset construction.

One interesting future direction is as follows. Recall that BindingNet only explicitly includes the atom-level information, and no quantum features are considered. Thus, one promising direction is to incorporate the quantum mechanisms into MD This challenge necessitates deeper integration of quantum physics knowledge into the model and algorithm design, a task we defer for future exploration.

Authors: Please allow a few lines extension during rebuttal. We will fix this in the final version.

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

REPRODUCIBILITY STATEMENT

In Sections 3 and 4, we have included the key modules of NeuralMD, and we list more details on the model architecture and hyperparameters in Appendix E. We will release the codes, checkpoints, and log files after the paper acceptance.

## A   RELATED WORKS AND PRELIMINARIES

**SE(3)-Equivariant Representation for Small Molecules and Proteins.** The molecular systems are indeed a set of atoms located in the 3D Euclidean space. From a machine learning point of view, the representation function or encoding function of such molecular systems needs to be group-symmetric, *i.e.*, the representation needs to be equivariant when we rotate or translate the whole system. Such symmetry is called the SE(3)-equivariance. Recently works (Liu et al., 2023a; Zhang et al., 2023) on molecules has provided a unified way of equivariant geometric modeling. They categorize the mainstream representation methods into three big venues: SE(3)-invariant models, SE(3)-equivariant models with spherical frame basis, and SE(3)-equivariant models with vector frame basis. (1) Invariant models that utilize invariant features (distances and angles) to predict the energies (Schütt et al., 2018; Klicpera et al., 2020), but the derived forces are challenging for ML optimization after integration. (2) Equivariant models with spherical frames that include a computationally expensive tensor-product operation (Smidt et al., 2018; Musaelian et al., 2022), which is unsuitable for large molecular systems. (3) Equivariant models with vector frames that have been explored for single stable molecules, including molecule representation and pretraining (Satorras et al., 2021; Schütt et al., 2021; Liu et al., 2023b), molecule conformation generation (Du et al., 2022), protein representation (Fan et al., 2022), and protein folding and design (Ingraham et al., 2019; Jumper et al., 2021). However, no one has tried it for binding complexes.

**Structure-Based Ligand Pose Generation in Equilibrium State.** One seemingly related but fundamentally different research line is the structure-based ligand pose generation. The core idea is to generate the ligand pose based on the protein structure. Statistical generative methods like autoregressive Peng et al. (2022) and denoising diffusion Corso et al. (2023); Guan et al. (2023) have been applied to this task. We want to highlight that these methods generate a trajectory, but they are essentially different for two reasons: (1) These works consider only the conformations at the equilibrium state, and no dynamics information is considered. (2) The inference of denoising diffusion generation is in the form of Monte Carlo (MC), which formulates a trajectory for each sampled sample. However, such trajectories include no physical information (*e.g.*, energy, and force), while the movements of molecules follow the Newtonian mechanics (*e.g.*, force fields) in MD simulation.

**Potential Energy and Force Learning for Molecular Dynamics Simulation.** One straightforward way of molecular dynamics (MD) simulation is through potential energy modeling. *Numerical methods* for MD simulation can be classified into classical MD and *ab-initio* MD, depending on using the classical mechanism or quantum mechanism to calculate the forces. Alternatively, a machine learning (ML) research line is to adopt geometric representation methods to learn the energies or the forces, *e.g.*, by the geometric methods listed above. The first work is DeePMD (Zhang et al., 2018), which targets learning the potential energy function at each conformation. For inference, the learned energy can be applied to update the atom placement using i-PI software Ceriotti (2014), composing the MD trajectories. TorchMD (Doerr et al., 2020) utilizes TorchMD-Net (Tholke & Fabritiis, 2022) for energy prediction, which will be fed into the velocity Verlet algorithm for MD simulation. Similarly, Musaelian et al. adopts Allegro (Musaelian et al., 2023a) model to learn the force at each conformation. The learned model will be used for MD trajectory simulation using LAMMPS Thompson et al. (2022). In theory, all the geometric models on small molecules Satorras et al. (2021); Schütt et al. (2021); Liu et al. (2023b) and proteins Fan et al. (2022) can be applied to the MD simulation task. However, there are two main challenges: (1) They require the interval between snapshots to be at the femtoseconds (1e-15 seconds) level for integration to effectively capture the motion of the molecules. (2) They take the position-energy pairs independently, and thus, they ignore their temporal correlations during learning.

**Trajectory Learning for Molecular Dynamics Estimation.** More recent works have explored MD simulation by directly learning the placement along the trajectories. There are two key differences

Table 4: Comparison of different numerical and machine learning (ML) methods for molecular dynamics (MD). AR for autoregressive and denoising for denoising diffusion method.

| Category | Method | Energy / Force Calculation | Dynamics | Objective Function | Publications |
|---|---|---|---|---|---|
| Numerical Methods | Classical MD | Classical Mechanics: Force Field | Newtonian Dynamics | – | – |
| | *Ab-initio* MD | Quantum Mechanics: DFT for Schrodinger Equation | Newtonian Dynamics | – | – |
| | Langevin MD | Classical Mechanics: Force Field | Langevin Dynamics | – | – |
| ML Methods | DeePMD (Zhang et al., 2018) | Atom-level Modeling | Newtonian Dynamics (i-PI) | Energy Prediction | PRL'18 |
| | TorchMD (Doerr et al., 2020) | Atom-level Modeling | Newtonian Dynamics (velocity Verlet) | Energy Prediction | ACS'20 |
| | Allegro-LAMMPS (Musaelian et al., 2023b) | Atom-level Modeling | Newtonian Dynamics (LAMMPS) | Force Prediction | ArXiv'23 |
| | **VerletMD (Ours, baseline)** | Atom-level Modeling | Newtonian Dynamics (velocity Verlet) | Energy Prediction | – |
| | CGDMS (Greener & Jones, 2021) | Atom-level Modeling | Newtonian Dynamics (velocity Verlet) | Position Prediction | PLOS'21 |
| | DiffMD (Wu & Li, 2023) | Atom-level Modeling | AR + Denoising | Position Prediction | AAAI'23 |
| | DFF (Arts et al., 2023) | Atom-level Modeling | AR + Denoising | Position Prediction | ACS'23 |
| | CG-MD (Fu et al., 2022b) | Atom-level Modeling | AR + Denoising | Position Prediction | TMLR'23 |
| | **LigandMD (Ours, baseline)** | Atom-level Modeling | AR + Denoising | Position Prediction | – |
| | **NeuralMD (Ours)** | Atom-level Modeling | Newtonian Dynamics (NODE) | Position Prediction | – |

between energy and trajectory prediction for MD: (1) Energy prediction takes each conformation and energy as IID, while trajectory learning optimizes the conformations along the whole trajectory, enforcing the temporal relation. (2) The time interval of trajectory learning is agnostic to the time interval, and energy prediction can be sensitive to longer MD simulations. More concretely, along such trajectory learning research line, CGDMS (Greener & Jones, 2021) builds an SE(3)-invariant model, followed by the velocity Verlet algorithm for MD simulation. DiffMD (Wu & Li, 2023) is a Markovian method and treats the dynamics between two consecutive snapshots as a coordinate denoising process. It then applies the SDE solver Song et al. (2020) to solve the molecular dynamics. A parallel work, DFF (Arts et al., 2023), applies a similar idea for MD simulation. CG-MD Fu et al. (2022b) encodes a hierarchical graph neural network model for an auto-regressive position generation and then adopts the denoising method for fine-tuning. However, these works disregard the prior knowledge of the Newtonian mechanics governing the motion of atoms.

**Protein-Ligand Binding Dynamics.** The MD simulation papers discussed so far are mainly for small molecules or proteins, not the binding dynamics. On the other hand, many works have studied the protein-ligand binding problem in the equilibrium state (Stepniewska-Dziubinska et al., 2018; Jiménez et al., 2018; Jones et al., 2021; Yang et al., 2023), but not the dynamics. In this work, we consider a more challenging task, which is the protein-ligand binding dynamics. The viability of this work is also attributed to the efforts of the scientific community, where more binding dynamics dataset has been revealed, including PLAS-5k Korlepara et al. (2022), MISATO Siebenmorgen et al. (2023), and PLAS-20k Priyakumar et al. (2023).

**Preliminary on Molecular Dynamics.** Molecular dynamics (MD) simulations predict how every atom in a molecular system moves over time, which is determined by the interatomic interactions following Newton's second law. Such a molecular system includes Small Molecules (Rapaport, 2004; Zhang et al., 2018), proteins (Karplus & Kuriyan, 2005; Frauenfelder et al., 2009; Arts et al., 2023), polymers (Fu et al., 2022b), and protein-ligand complexes (Korlepara et al., 2022; Siebenmorgen et al., 2023). Typically, an MD simulation is composed of two main steps, *i.e.*, (1) the energy and force calculation and (2) integration of the equations of motion governed by Newton's second law of motion, using the initial conditions and forces calculated in step (1). As the initial condition, the initial positions and velocities are given for all the particles (*e.g.*, atoms) in the molecular system; the MD simulation repeats the two steps to get a trajectory. Such MD simulations can be used to calculate the equilibrium and transport properties of molecules, materials, and biomolecular systems (Frenkel & Smit, 2002).

In such an MD simulation, one key factor is estimating the forces on each atom. The function that gives the energy of a molecular system as a function of its structure (and forces via the gradient of the energy with respect to those atomic coordinates) is referred to as a potential energy surface (PES). In general, MD simulations integrate the equations of motion using a PES from one of two sources: (1) Classical MD using the force fields, which are parameterized equations that approximate the true PES, and are less costly to evaluate, allowing for the treatment of larger systems and longer timesteps. (2) *ab-inito* MD (which calculates the energy of a molecular system via electronic structure methods, *e.g.*, DFT) provide more accurate PES, but are limited in the system size and timesteps that are practically accessible due to the cost of evaluating the PES at a given point.

## B  GROUP SYMMETRY AND EQUIVARIANCE

In this article, a 3D molecular graph is represented by a collection of 3D point clouds. The corresponding symmetry group is SE(3), which consists of translations and rotations. Recall that we define the notion of equivariance functions in $\mathbf{R}^3$ in the main text through group actions. Formally, the group SE(3) is said to act on $\mathbf{R}^3$ if there is a mapping $\phi : \text{SE}(3) \times \mathbf{R}^3 \to \mathbf{R}^3$ satisfying the following two conditions:

1. if $e \in \text{SE}(3)$ is the identity element, then

$$\phi(e, \boldsymbol{r}) = \boldsymbol{r} \quad \text{for } \forall \boldsymbol{r} \in \mathbf{R}^3.$$

2. if $g_1, g_2 \in \text{SE}(3)$, then

$$\phi(g_1, \phi(g_2, \boldsymbol{r})) = \phi(g_1 g_2, \boldsymbol{r}) \quad \text{for } \forall \boldsymbol{r} \in \mathbf{R}^3.$$

Then, there is a natural SE(3) action on vectors $\boldsymbol{r}$ in $\mathbf{R}^3$ by translating $\boldsymbol{r}$ and rotating $\boldsymbol{r}$ for multiple times. For $g \in \text{SE}(3)$ and $\boldsymbol{r} \in \mathbf{R}^3$, we denote this action by $g\boldsymbol{r}$. Once the notion of group action is defined, we say a function $f : \mathbf{R}^3 \to \mathbf{R}^3$ that transforms $\boldsymbol{r} \in \mathbf{R}^3$ is equivariant if:

$$f(g\boldsymbol{r}) = gf(\boldsymbol{r}), \quad \text{for } \forall \ \boldsymbol{r} \in \mathbf{R}^3.$$

On the other hand, $f : \mathbf{R}^3 \to \mathbf{R}^1$ is invariant, if $f$ is independent of the group actions:

$$f(g\boldsymbol{r}) = f(\boldsymbol{r}), \quad \text{for } \forall \ \boldsymbol{r} \in \mathbf{R}^3.$$

For some scenarios, our problem is chiral sensitive. That is, after mirror reflecting a 3D molecule, the properties of the molecule may change dramatically. In these cases, it's crucial to include reflection transformations into consideration. More precisely, we say an SE(3) equivariant function $f$ is **reflection anti-symmetric**, if:

$$f(\rho \boldsymbol{r}) \neq f(\boldsymbol{r}), \tag{9}$$

for reflection $\rho \in \text{E}(3)$.

## C  Equivariant Modeling With Vector Frames

**Frame** is a popular terminology in science areas. In physics, the frame is equivalent to a coordinate system. For example, we may assign a frame to all observers, although different observers may collect different data under different frames, the underlying physics law should be the same. In other words, denote the physics law by $f$, then $f$ should be an equivariant function.

There are certain ways to choose the frame basis, and below we introduce two main types: the orthogonal basis and the protein backbone basis. The orthogonal basis can be built for flexible 3D point clouds such as atoms, while the protein backbone basis is specifically proposed to capture the protein backbone.

### C.1  Basis

Since there are three orthogonal directions in $\mathbf{R}^3$, one natural frame in $\mathbf{R}^3$ can be a frame consisting of three orthogonal vectors:

$$F = (\boldsymbol{e}_1, \boldsymbol{e}_2, \boldsymbol{e}_3).$$

Once equipped with a frame (coordinate system), we can project all geometric quantities to this frame. For example, an abstract vector $\boldsymbol{x} \in \mathbf{R}^3$ can be written as $\boldsymbol{x} = (r_1, r_2, r_3)$ under the frame $F$, if: $\boldsymbol{x} = r_1\boldsymbol{e}_1 + r_2\boldsymbol{e}_2 + r_3\boldsymbol{e}_3$. A vector frame further requires the three orthonormal vectors in $(\boldsymbol{e}_1, \boldsymbol{e}_2, \boldsymbol{e}_3)$ to be equivariant. Intuitively, a vector frame will transform according to the global rotation or translation of the whole system. Once equipped with a vector frame, we can project vectors into this frame in an equivariant way:

$$\boldsymbol{x} = \tilde{r}_1\boldsymbol{e}_1 + \tilde{r}_2\boldsymbol{e}_2 + \tilde{r}_3\boldsymbol{e}_3. \tag{10}$$

We call the process of $\boldsymbol{x} \to \tilde{r} := (\tilde{r}_1, \tilde{r}_2, \tilde{r}_3)$ the **scalarization** or **projection** operation. Since $\tilde{r}_i = \boldsymbol{e}_i \cdot \boldsymbol{x}$ is expressed as an inner product between vector vectors, we know that $\tilde{r}$ consists of scalars.

In this article, we assign a vector frame to each node/edge, therefore we call them the local frames. We note that, in this section, we prove the equivariance property of the vector frame basis using the Gram-Schmidt methods. However, similar equivariance property can be easily guaranteed for the vector frame bases in Section 3 after we remove the mass center of the molecular system.

In the main body, we constructed three vector frames based on three granularities. Here we provide the proof on the protein backbone frame. Say the three backbone atoms in on proteins are $\boldsymbol{x}_i, \boldsymbol{x}_j, \boldsymbol{x}_k$ respectively. Then the vector frame is defined by:

$$\text{Vector-Frame}(\boldsymbol{x}_i, \boldsymbol{x}_j) := \textbf{Gram-Schmidt}\{\boldsymbol{x}_i - \boldsymbol{x}_j, \boldsymbol{x}_i - \boldsymbol{x}_k, (\boldsymbol{x}_i - \boldsymbol{x}_j) \times (\boldsymbol{x}_i - \boldsymbol{x}_k)\}. \tag{11}$$

The Gram-Schmidt orthogonalization makes sure that the Vector-Frame$(\boldsymbol{x}_i, \boldsymbol{x}_j)$ is orthonormal.

**Reflection Antisymmetric**  Since we implement the cross product $\times$ for building the local frames, the third vector in the frame is a pseudo-vector. Then, the **projection** operation is not invariant under reflections (the inner product between a vector and a pseudo-vector change signs under reflection). Therefore, our model can discriminate two 3D geometries with different chiralities.

Our local frames also enable us to output vectors equivariantly by multiplying scalars $(v_1, v_2, v_3)$ with the frame: $\boldsymbol{v} = v_1 \cdot \boldsymbol{e}_1 + v_2 \cdot \boldsymbol{e}_2 + v_3 \cdot \boldsymbol{e}_3$.

---

**Equivariance w.r.t. cross-product**  The goal is to prove that the cross-product is equivariant to the SE(3)-group, *i.e.*:

$$gx \times gy = g(x \times y), \qquad g \in \text{SE(3)-Group} \tag{12}$$

---

**Geometric proof.** From intuition, with rotation matrix $g$, we are transforming the whole basis, thus the direction of $gx \times gy$ changes equivalently with $g$. And for the value/length of $gx \times gy$, because $|gx \times gy| = \|gx\| \cdot \|gy\| \cdot \sin\theta = \|x\| \cdot \|y\| \cdot \sin\theta = |x \times y|$. So the length stays the same, and the direction changes equivalently. Intuitively, this interpretation is quite straightforward.

**Analytical proof.** A more rigorous proof can be found below:

*Proof.* First, we have that for the rotation matrix $g$:

$$gx \times gy = \begin{bmatrix} \boldsymbol{g}_1^T \boldsymbol{x} \\ \boldsymbol{g}_2^T \boldsymbol{x} \\ \boldsymbol{g}_3^T \boldsymbol{x} \end{bmatrix} \times \begin{bmatrix} \boldsymbol{g}_1^T \boldsymbol{y} \\ \boldsymbol{g}_2^T \boldsymbol{y} \\ \boldsymbol{g}_3^T \boldsymbol{y} \end{bmatrix} = \begin{bmatrix} \boldsymbol{g}_2^T \boldsymbol{x} \cdot \boldsymbol{g}_3^T \boldsymbol{y} - \boldsymbol{g}_3^T \boldsymbol{x} \cdot \boldsymbol{g}_2^T \boldsymbol{y} \\ -\boldsymbol{g}_1^T \boldsymbol{x} \cdot \boldsymbol{g}_3^T \boldsymbol{y} + \boldsymbol{g}_3^T \boldsymbol{x} \cdot \boldsymbol{g}_1^T \boldsymbol{y} \\ \boldsymbol{g}_1^T \boldsymbol{x} \cdot \boldsymbol{g}_2^T \boldsymbol{y} - \boldsymbol{g}_2^T \boldsymbol{x} \cdot \boldsymbol{g}_1^T \boldsymbol{y} \end{bmatrix}, \tag{13}$$

where $\boldsymbol{g}_i, \boldsymbol{x}, \boldsymbol{y} \in \mathbb{R}^{3 \times 1}$.

Because $A^T C \cdot B^T D - A^T D \cdot B^T C = (A \times B)^T (C \times D)$, so we can have:

$$gx \times gy = \begin{bmatrix} \boldsymbol{g}_2^T \boldsymbol{x} \cdot \boldsymbol{g}_3^T \boldsymbol{y} - \boldsymbol{g}_3^T \boldsymbol{x} \cdot \boldsymbol{g}_2^T \boldsymbol{y} \\ -\boldsymbol{g}_1^T \boldsymbol{x} \cdot \boldsymbol{g}_3^T \boldsymbol{y} + \boldsymbol{g}_3^T \boldsymbol{x} \cdot \boldsymbol{g}_1^T \boldsymbol{y} \\ \boldsymbol{g}_1^T \boldsymbol{x} \cdot \boldsymbol{g}_2^T \boldsymbol{y} - \boldsymbol{g}_2^T \boldsymbol{x} \cdot \boldsymbol{g}_1^T \boldsymbol{y} \end{bmatrix} = \begin{bmatrix} (\boldsymbol{g}_2 \times \boldsymbol{g}_3)^T (\boldsymbol{x} \times \boldsymbol{y}) \\ (\boldsymbol{g}_3 \times \boldsymbol{g}_1)^T (\boldsymbol{x} \times \boldsymbol{y}) \\ (\boldsymbol{g}_1 \times \boldsymbol{g}_2)^T (\boldsymbol{x} \times \boldsymbol{y}). \end{bmatrix} \tag{14}$$

Then because:

$$\det(g) = (\boldsymbol{g}_2 \times \boldsymbol{g}_3)^T \boldsymbol{g}_1 = \boldsymbol{g}_1^T \boldsymbol{g}_1 = 1$$
$$\Longrightarrow (\boldsymbol{g}_2 \times \boldsymbol{g}_3)^T \boldsymbol{g}_1 \boldsymbol{g}_1^{-1} = \boldsymbol{g}_1^T \boldsymbol{g}_1 \boldsymbol{g}_1^{-1} \tag{15}$$
$$\Longrightarrow (\boldsymbol{g}_2 \times \boldsymbol{g}_3)^T = \boldsymbol{g}_1^T.$$

Thus, we can have

$$gx \times gy = \begin{bmatrix} (\boldsymbol{g}_2 \times \boldsymbol{g}_3)^T (\boldsymbol{x} \times \boldsymbol{y}) \\ (\boldsymbol{g}_3 \times \boldsymbol{g}_1)^T (\boldsymbol{x} \times \boldsymbol{y}) \\ (\boldsymbol{g}_1 \times \boldsymbol{g}_2)^T (\boldsymbol{x} \times \boldsymbol{y}) \end{bmatrix} = \begin{bmatrix} \boldsymbol{g}_1^T (\boldsymbol{x} \times \boldsymbol{y}) \\ \boldsymbol{g}_2^T (\boldsymbol{x} \times \boldsymbol{y}) \\ \boldsymbol{g}_3^T (\boldsymbol{x} \times \boldsymbol{y}) \end{bmatrix} = g(\boldsymbol{x} \times \boldsymbol{y}). \tag{16}$$

$\square$

---

**Rotation symmetric**   The goal is to prove

$$\text{Vector-Frame}(g\boldsymbol{x}_i, g\boldsymbol{x}_j) = g\textbf{Gram-Schmidt}\{\boldsymbol{x}_i - \boldsymbol{x}_j, \boldsymbol{x}_i - \boldsymbol{x}_k, (\boldsymbol{x}_i - \boldsymbol{x}_j) \times (\boldsymbol{x}_i - \boldsymbol{x}_k)\}. \tag{17}$$

---

*Proof.* Thus we can have:

$$\begin{aligned} \text{Vector-Frame}(g\boldsymbol{x}_i, g\boldsymbol{x}_j) &= \textbf{Gram-Schmidt}\{g\boldsymbol{x}_i - g\boldsymbol{x}_j, g\boldsymbol{x}_i - g\boldsymbol{x}_k, (g\boldsymbol{x}_i - g\boldsymbol{x}_j) \times (g\boldsymbol{x}_i - g\boldsymbol{x}_k)\} \\ &= \textbf{Gram-Schmidt}\{g(\boldsymbol{x}_i - \boldsymbol{x}_j), g(\boldsymbol{x}_i - \boldsymbol{x}_k), g((\boldsymbol{x}_i - \boldsymbol{x}_j) \times (\boldsymbol{x}_i - \boldsymbol{x}_k))\}. \end{aligned} \tag{18}$$

Recall that Gram-Schmidt projection (**Gram-Schmidt**$\{\boldsymbol{v}_1, \boldsymbol{v}_2, \boldsymbol{v}_3\}$) is:

$$\begin{aligned} \boldsymbol{u}_1 &= \boldsymbol{v}_1, & \boldsymbol{e}_1 &= \frac{\boldsymbol{v}_1}{\|\boldsymbol{v}_1\|}, \\ \boldsymbol{u}_2 &= \boldsymbol{v}_2 - \frac{\boldsymbol{u}_1^T \boldsymbol{v}_2}{\|\boldsymbol{u}_1\|} \boldsymbol{u}_1, & \boldsymbol{e}_2 &= \frac{\boldsymbol{v}_2}{\|\boldsymbol{v}_2\|}, \\ \boldsymbol{u}_3 &= \boldsymbol{v}_3 - \frac{\boldsymbol{u}_1^T \boldsymbol{v}_3}{\|\boldsymbol{u}_1\|} \boldsymbol{u}_1 - \frac{\boldsymbol{u}_2^T \boldsymbol{v}_3}{\|\boldsymbol{u}_2\|} \boldsymbol{u}_2, & \boldsymbol{e}_3 &= \frac{\boldsymbol{v}_3}{\|\boldsymbol{v}_3\|}. \end{aligned} \tag{19}$$

Thus, the Gram-Schmidt projection on the rotated vector (**Gram-Schmidt**$\{g\boldsymbol{v}_1, g\boldsymbol{v}_2, g\boldsymbol{v}_3\}$) is:

$$\begin{aligned} \boldsymbol{u}_1' &= g\boldsymbol{v}_1, \\ \boldsymbol{u}_2' &= g\boldsymbol{v}_2 - g\frac{\boldsymbol{u}_1^T \boldsymbol{v}_2}{\|\boldsymbol{u}_1\|} \boldsymbol{u}_1, \\ \boldsymbol{u}_3' &= g\boldsymbol{v}_3 - g\frac{\boldsymbol{u}_1^T \boldsymbol{v}_3}{\|\boldsymbol{u}_1\|} \boldsymbol{u}_1 - g\frac{\boldsymbol{u}_2^T \boldsymbol{v}_3}{\|\boldsymbol{u}_2\|} \boldsymbol{u}_2, \end{aligned} \tag{20}$$

Thus, **Gram-Schmidt**$\{g\boldsymbol{v}_1, g\boldsymbol{v}_2, g\boldsymbol{v}_3\} = g\textbf{Gram-Schmidt}\{\boldsymbol{v}_1, \boldsymbol{v}_2, \boldsymbol{v}_3\}$.

$\square$

**Transition symmetric**

$$\text{Vector-Frame}(\boldsymbol{x}_i + \delta\boldsymbol{x}, \boldsymbol{x}_j + \delta\boldsymbol{x}) = \textbf{Gram-Schmidt}\{\boldsymbol{x}_i - \boldsymbol{x}_j, \boldsymbol{x}_i - \boldsymbol{x}_k, (\boldsymbol{x}_i - \boldsymbol{x}_j) \times (\boldsymbol{x}_i - \boldsymbol{x}_k)\}. \quad (21)$$

*Proof.* Because the basis is based on the difference of coordinates, it is straightforward to observe that $\textbf{Gram-Schmidt}\{\boldsymbol{v}_1 + \boldsymbol{t}, \boldsymbol{v}_2 + \boldsymbol{t}, \boldsymbol{v}_3 + \boldsymbol{t}\} = \textbf{Gram-Schmidt}\{\boldsymbol{v}_1, \boldsymbol{v}_2, \boldsymbol{v}_3\}$. So the frame operation is transition equivariant. We also want to highlight that for all the other vector frame bases introduced in Section 3, we remove the mass center for each molecular system, thus, we can guarantee the transition equivariance property. □

**Reflection antisymmetric**

$$\text{Vector-Frame}(\boldsymbol{x}_i, \boldsymbol{x}_j) \neq \text{Vector-Frame}(-\boldsymbol{x}_i, -\boldsymbol{x}_j). \quad (22)$$

*Proof.* From intuition, this makes sense because the cross-product is anti-symmetric.

A simple counter-example is the original geometry $R$ and the reflected geometry by the original point $-R$. Thus the two bases before and after the reflection group is the following:

$$\textbf{Gram-Schmidt}\{\boldsymbol{x}_i - \boldsymbol{x}_j, \boldsymbol{x}_i - \boldsymbol{x}_k, (\boldsymbol{x}_i - \boldsymbol{x}_j) \times (\boldsymbol{x}_i - \boldsymbol{x}_k)\} \quad (23)$$

$$\textbf{Gram-Schmidt}\{-\boldsymbol{x}_i + \boldsymbol{x}_j, -\boldsymbol{x}_i + \boldsymbol{x}_k, (\boldsymbol{x}_i - \boldsymbol{x}_j) \times (\boldsymbol{x}_i - \boldsymbol{x}_k)\}. \quad (24)$$

The bases between $\boldsymbol{v}_1, \boldsymbol{v}_2, \boldsymbol{v}_3$ and $\{-\boldsymbol{v}_1, -\boldsymbol{v}_2, \boldsymbol{v}_3\}\}$ are different, thus such frame construction is reflection anti-symmetric.

□

If you are able to get the above derivations carefully with a good understanding, then you can tell that this can be trivially generalized to arbitrary vector frames.

## C.2 SCALARIZATION

Once we have the three vectors as the vector frame basis, the next thing is to do modeling. Suppose the frame is $\mathcal{F} = (\boldsymbol{e}_1, \boldsymbol{e}_2, \boldsymbol{e}_3)$, then for a vector (tensor) $\boldsymbol{h}$, the scalarization is:

$$\boldsymbol{h} \odot \mathcal{F} = (\boldsymbol{h} \odot \boldsymbol{e}_1, \boldsymbol{h} \odot \boldsymbol{e}_2, \boldsymbol{h} \odot \boldsymbol{e}_3) = (\boldsymbol{h}_1, \boldsymbol{h}_2, \boldsymbol{h}_3). \quad (25)$$

# D  SPECIFICATIONS ON MISATO

In this section, we provide more details on the MISATO dataset (Siebenmorgen et al., 2023).
For small molecule ligands, we ignore the Hydrogen atoms.

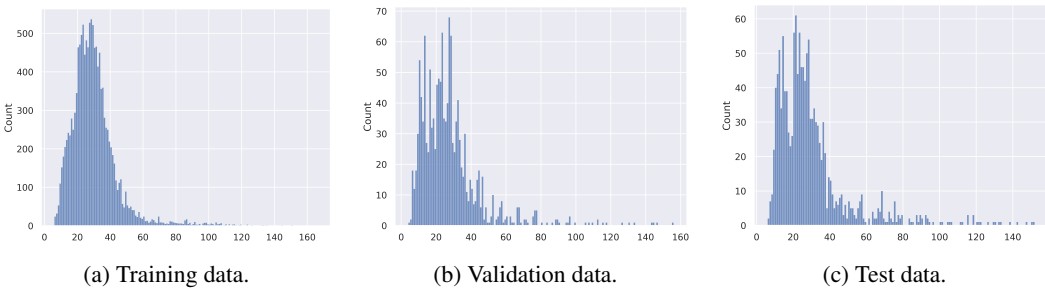

(a) Training data.            (b) Validation data.            (c) Test data.

Figure 5: Distribution on # atoms in small molecule ligands for all protein-ligand complex.

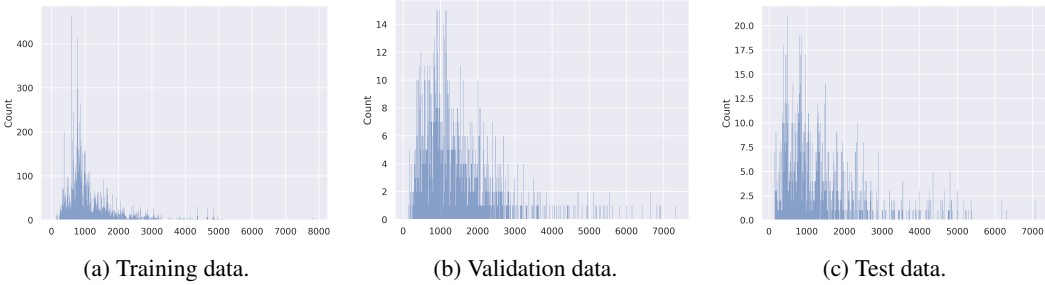

(a) Training data.            (b) Validation data.            (c) Test data.

Figure 6: Distribution on # residues in proteins for all protein-ligand complex.

We also plot the distribution of the energy gap between each time step and the initial snapshot, *i.e.*,
$E_t - E_0$. The distribution is in Figure 7. We can observe that as the time processes, the mean of the
energy stays almost the same, yet the variance gets higher.

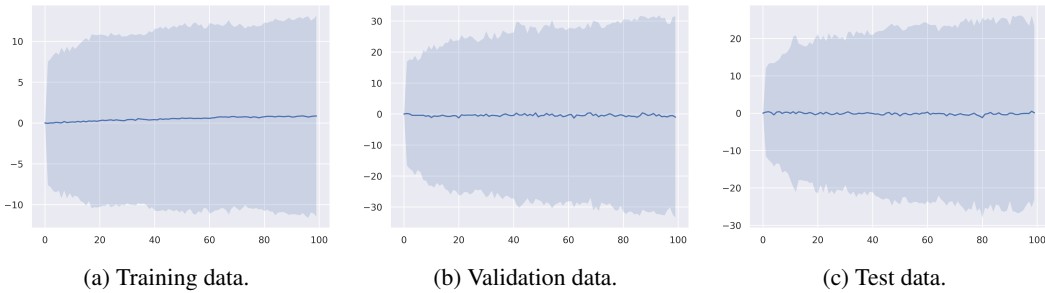

(a) Training data.            (b) Validation data.            (c) Test data.

Figure 7: Distribution on energy $E_t - E_0$.

# E DETAILS OF NEURALMD

In this section, we provide more details on the model architecture in Figure 8, and hyperparameter details in Table 5.

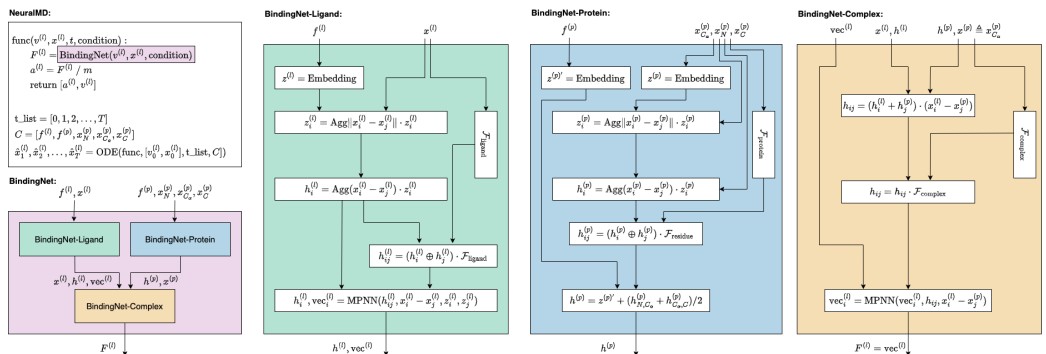

Figure 8: Detailed pipeline of NeuralMD. In the three key modules of BindingNet, there are three vertical boxes, corresponding to three granularities of vector frames, as in Equations (2) to (4).

Here, we explain each of the three modules in detail and list the dimensions of each variable to make it easier for readers to understand. Suppose the representation dimension is $d$.

**BindingNet-Ligand:**

- $z^{(l)} = \text{Embedding}(f^{(l)}) \in \mathbb{R}^{N_{\text{atom}} \cdot d}$ is atom type embedding.
- Then for each atom type embedding $z^{(l)}$, we add a normalization by multiplying it with the RBF of distance among the neighborhoods, and the resulting atom type embedding stays the same dimension $z^{(l)} \in \mathbb{R}^{N_{\text{atom}} \times d}$.
- $\{h_i^{(l)} = \text{Agg}_j(x_i^{(l)} - x_j^{(l)}) \cdot z_i^{(l)}\} \in \mathbb{R}^{N_{\text{atom}} \cdot d \cdot 3}$ is the equivariant representation of each atom.
- $\{h_{ij}^{(l)}\} \in \mathbb{R}^{N_{\text{edge}} \cdot 2d \cdot 3}$ is the invariant representation after scalarization. Then we will take a simple sum-pooling, followed by an MLP to get the invariant representation $h_{ij}^{(l)} \in \mathbb{R}^{N_{\text{edge}} \cdot d}$.
- Finally, we will repeat $L$ layers of MPNN:

$$\text{vec}_i^{(l)} = \text{vec}_i^{(l)} + \text{Agg}_j\big(\text{vec}_i^{(l)} \cdot \text{MLP}(h_{ij}) + (x_i^{(l)} - x_j^{(p)}) \cdot \text{MLP}(h_{ij})\big), \quad //\{\text{vec}_i^{(l)}\} \in \mathbb{R}^{N_{\text{atom}} \cdot 3}$$
$$h_i^{(l)} = h_i^{(l)} + \text{Agg}_j\big(\text{MLP}(h_{ij})\big). \quad //\{h_i^{(l)}\} \in \mathbb{R}^{N_{\text{atom}} \cdot d}$$

$$(26)$$

**BindingNet-Protein:**

- $z^{(p)} \in \mathbb{R}^{N_{\text{backbone-atom}} \cdot d}$ is the backbone-atom type representation by aggregating the neighbors without the cutoff $c$.
- $\tilde{z}^{(p)} \in \mathbb{R}^{N_{\text{backbone-atom}} \cdot d}$ is the backbone-atom type representation.
- $\{h_i^{(p)}\} \in \mathbb{R}^{N_{\text{backbone-atom}} \cdot d \cdot 3}$ is the backbone-atom equivariant representation.
- $\{h_{ij}^{(p)}\} \in \mathbb{R}^{N_{\text{edge}} \cdot 2d \cdot 3}$ is the invariant representation after scalarization. Then we take a simple sum-pooling, followed by an MLP to get the invariant representation $\{h_{ij}^{(l)}\} \in \mathbb{R}^{N_{\text{edge}} \cdot d}$.
- Finally, we get the residue-level representation as $h^{(p)} = \tilde{z}^{(p)} + (h_{N,C_\alpha}^{(p)} + h_{C_\alpha,C}^{(p)})/2 \in \mathbb{R}^{N_{\text{residue}} \cdot d}$.

**BindingNet-Complex:**

- $\{h_{ij}\} \in \mathbb{R}^{N_{\text{edge}} \cdot d \cdot 3}$ is the equivariant interaction/edge representation.
- $\{h_{ij} = h_{ij} \cdot \mathcal{F}_{\text{complex}}\}^{\mathbb{R}^{N_{\text{edge}} \cdot d \cdot 3}}$ is the scalarization. Then we take a simple sum-pooling, followed by an MLP to get the invariant representation $\{h_{ij}^{(l)}\} \in \mathbb{R}^{N_{\text{edge}} \cdot d}$.
- The final output is obtained by $L$ MPNN layers as:

$$\text{vec}_{ij}^{(pl)} = \text{vec}_i^{(l)} \cdot \text{MLP}(h_{ij}) + (x_i^{(l)} - x_j^{(p)}) \cdot \text{MLP}(h_{ij}), \quad //\{\text{vec}_{ij}^{(pl)}\} \in \mathbb{R}^{N_{\text{edge}} \cdot 3}$$
$$F_i^{(l)} = \text{vec}_i^{(l)} + \text{Agg}_{j \in \mathcal{N}(i)}\text{vec}_{ij}^{(pl)}. \quad //\{F_i^{(l)}\} \in \mathbb{R}^{N_{\text{atom}} \cdot 3}$$

$$(27)$$

Table 5: Hyperparameter specifications for NeuralMD.

| Hyperparameter | Value |
|---|---|
| # layers | {5} |
| cutoff $c$ | {5} |
| learning rate | {1e-4, 1e-3} |
| optimizer | {SGD, Adam } |

# F ABLATION STUDIES: FLEXIBLE BINDING

Recall that, in the main paper, we have discussed using the *semi-flexible* binding setting, *i.e.*, proteins with rigid structures while small molecule ligands with flexible structures, and the goal is to predict the trajectories of the ligands. As discussed in Section 5, if we want to take both proteins and ligands with flexible structures, one limitation is the GPU memory cost. However, we would like to mention that it is possible to do NeuralMD on protein-ligand with small volume, and we take an ablation study to test them as below.

**Problem Setup.** Both the proteins and ligands are flexible, and we want to predict their trajectories simultaneously. In the main paper, we consider three levels of vector frames. Here in the flexible setting, due to the large volume of atoms in the protein-ligand complex, we are only able to consider two levels, *i.e.*, the atom-level and residue-level. Thus, the backbone model (BindingNet) also changes accordingly. The performance is shown in Table 6, and we can see that NeuralMD is consistently better than the GNN-MD on all three metrics and all 10 single trajectories. We omit the multi-trajectory experiments due to the memory limitation.

Table 6: Results on ten single-trajectory binding dynamics prediction. Results with optimal training loss are reported. Four evaluation metrics are considered: MAE (Å, ↓), MSE (↓), and Stability (%, ↓).

|  |  | GNN-MD | NeuralMD-ODE (Ours) |
|---|---|---|---|
| 5WIJ | MAE | 7.126 | **3.101** |
|  | MSE | 4.992 | **2.070** |
|  | Stability | 68.317 | **30.655** |
| 4ZX0 | MAE | 9.419 | **2.580** |
|  | MSE | 6.269 | **1.724** |
|  | Stability | 67.492 | **29.013** |
| 3EOV | MAE | 10.695 | **3.664** |
|  | MSE | 7.447 | **2.521** |
|  | Stability | 67.782 | **39.714** |
| 4K6W | MAE | 8.347 | **3.056** |
|  | MSE | 5.605 | **2.007** |
|  | Stability | 63.839 | **36.972** |
| 1KTI | MAE | 8.900 | **6.815** |
|  | MSE | 5.820 | **4.268** |
|  | Stability | 65.010 | **26.805** |
| 1XP6 | MAE | 8.496 | **1.910** |
|  | MSE | 5.673 | **1.276** |
|  | Stability | 70.019 | **33.907** |
| 4YUR | MAE | 11.710 | **7.568** |
|  | MSE | 7.759 | **4.966** |
|  | Stability | 69.163 | **34.636** |
| 4G3E | MAE | 1314.425 | **3.282** |
|  | MSE | 814.641 | **2.152** |
|  | Stability | 65.703 | **21.095** |
| 6B7F | MAE | 182.278 | **3.166** |
|  | MSE | 115.688 | **2.121** |
|  | Stability | 72.027 | **26.931** |
| 3B9S | MAE | 3.590 | **2.477** |
|  | MSE | 2.431 | **1.615** |
|  | Stability | 54.890 | **18.817** |

## G LANGEVIN DYNAMICS

The **Langevin dynamics** is defined as

$$m\boldsymbol{a} = -\nabla U(\boldsymbol{x}) - \gamma m \boldsymbol{v} + \sqrt{2m\gamma k_B T}R(t), \tag{28}$$

where $\gamma$ is the damping constant or collision frequency, $T$ is the temperature, $k_B$ is the Boltzmann's constant, and $R(t)$ is a delta-correlated stationary Gaussian process with zero-mean.

The term $m\boldsymbol{a}$ is called inertial force. In **overdamped Langevin dynamics** (a.k.a. Brownian dynamics), the inertial force is much smaller than the other three terms, thus it is considered negligible. Then the equation for overdamped Langevin dynamics is:

$$-\nabla U(\boldsymbol{x}) - \gamma m \boldsymbol{v} + \sqrt{2m\gamma k_B T}R(t) = 0. \tag{29}$$

Thus, the trajectories are given by:

$$\begin{aligned}
\boldsymbol{x}_{t+1} - \boldsymbol{x}_t &= -\frac{1}{\gamma m}\nabla U(x) + \frac{\sqrt{2m\gamma k_B T}}{\gamma m}R(t) \\
&= -\frac{D}{k_B T}\nabla U(X) + \sqrt{2D}R(t),
\end{aligned} \tag{30}$$

where $D = k_B T/\gamma$. However, such an overdamped Langevin dynamics does not hold for small particles like molecules. Because the assumption that inertial force can be ignored is only valid for large particles.

