# OpenReview forum: "A Multi-Grained Group Symmetric Framework for Learning Protein-Ligand Binding Dynamics"
_ICLR.cc/2024/Conference — Submitted to ICLR 2024_

### Official Review · Reviewer_oR9H · 2023-10-24

**Soundness:** 1 poor
**Presentation:** 2 fair
**Contribution:** 1 poor
**Rating:** 3
**Confidence:** 4

**Summary:**

This paper aims to learn a 2nd order neural ODE which emulates molecular dynamics trajectories of ligand-protein interactions from the Misato dataset. The force prediction is done via a new architecture called BindingNet, which is based on 3 levels of frames (ligand, protein atom, protein residue) and which is claimed to be SE(3)-equivariant. The method is shown to be more accurate than diffusion-based next-step predictors and more stable than simulating the dynamics with a learned force field.

**Strengths:**

* It is encouraging to see the first works make use of the MISATO dataset, which contains a wealth of information about the very difficult problem of protein-ligand binding.
* Directly learning the long-timescale dynamics with a neural ODE rather than a fixed-timestep next-step predictor is an interesting idea which merits broader exploration.

**Weaknesses:**

* Problem formulation. Although learning a neural ODE for long-timescale dynamics is an interesting idea, the deterministic problem formulation seems inappropriate as MD simulation itself is not necessarily deterministic. Thermostats and barostats typically introduce stochasticity, and even in their absence the removal of water molecules (which are explicit in the MISATO simulations) injects intrinsic uncertainty into the modeling problem. Hence, the dynamics are fundamentally stochastic and I am skeptical that any neural ODE can faithfully capture the long-timescale dynamics of these protein-ligand systems.

* Performance and baselines. Even in the single-trajectory setting, the MAE for the ligand coordinates seems rather large, in the range of 2-6 angstroms. This is perhaps due to the suboptimal problem formulation already discussed. However, the numbers provided are not meaningful because we do not know what the RMSF is in these simulations. The RMSF is the best result that can be achieved by a single static structure and is an essential missing baseline. The MAE in the multi-trajectory setting is even larger (7 anstroms) and suggests a complete dissociation of the ligand from the binding pocket. Finally, the stability metrics are only marginally better than DenoisingMD and do not represent a qualitative resolution of the problem.

* Misleading title. “Group symmetric” suggests a much more general framework than SE(3)-equivariance, and the key contribution of the paper is not the equivariant architecture in my opinion.

* Mathematical errors. Although the paper places much emphasis on SE(3)-equivariance, the construction of the atom-level and residue-level frames appears to be non-equivariant. Specifically, the cross product is not translation equivariant since
$$(x_i + t) \times (x_j + t) = (x_i \times x_j) +  (x_i - x_j) \times t$$
is not a function of $(x_t\times x_j)$ and $t$. Noticeably, the appendix claims the cross-product to be SE(3)-equivariant but only establishes rotation equivariance. Further, while I agree that the local frame construction described in the appendix is equivariant, this is different from what is done in the main text, since there is no nearest-neighbor atom $x_k$ and absolute positions $x_i, x_j$ are used.

Justification for score. Although the paper is a commendable attempt to learn from MD data, I am not convinced that the problem formulation makes sense, and the experimental results are rather weak. The frame based architecture is not a significant technical advancement over existing frame architectures and appears to have flaws.

**Questions:**

* The architectural details in Eq 5 and Fig 4 are very unclear, with several ambiguous uses of the dot product. Please label the embeddings with the dimensionality (so it is clear if they are scalars or vectors) and define symbols before they are used (for example $h_i$). It is also not clear what kind of MPNN is used and what is the meaning of the number of "layers."
* Please clarify if by MAE you mean RMSD or some alternative definition of positional error (with or without a factor of $\sqrt{3}$)
* "We keep the same backbone model (BindingNet) for energy or force prediction for all the baselines." If so, please provide significantly more details about how these baselines were retrained.
* If this work considers the semi-flexible, how did you deal with the protein movement that is present in the MISATO MD trajectories?
* What is your integration timestep? If it is adaptive, please provide more details about how many timesteps are required for a typical 8ns simulation.
* In the single-trajectory setting, how is the temporal division carried out?

---

> ### Author Response · Authors · 2023-11-16
> **Response (1/3)**
>
> Thank you for raising your comments. There is one critical comment and some misunderstandings, and we hope we have addressed them in our rebuttal. If there are any other follow-up questions, we will be happy to address them.
>
>
> **Math is correct. BindingNet is SE(3)-equivariant because the mass center is removed.**
>
> Thank you for raising this concern. However, we mentioned that all the atom and backbone residue positions are reduced by the mass point of the molecular system in the first version, thus it is translation equivariance. We have described this in **line 3 of Algorithm 1**. Further for clarification, we have highlighted this in the revised version, especially in the proof (Sec C).
>
>
> **Correction on problem formulation**
>
> Thank you for raising this question.
> - First on the solvent molecules. The ideal case is that the MISATO authors can provide them in the dataset. We contacted the authors and tried to extract the dynamics of solvent molecules, but such information is still missing in the dataset (we are waiting for the replies now).
> - However, such missing information can be easily captured using our NeuralMD framework.
>     - Current in Newtonian dynamics: $F = \text{BindingNet(protein-ligand)}$
>     - Add random terms to mimic solvent molecules as Langevin dynamics: $F = \text{BindingNet(protein-ligand)} - \gamma m v + \sqrt{2m \gamma k_BT} R(t)$. We have this explained in the revised manuscript (the colored text at the end of Section 3.3), please feel free to check. For the rest part of the manuscript (e.g., Abstract and Sec 1), we can change it from ODE to SDE easily in the final version.
> - The results of *single trajectory* and *multiple trajectories* are attached below. You can also find them in the revised manuscript.
> - MISATO is under the NVT configuration, and it uses thermostats, then Langevin dynamics can capture such stochasticity [1].
>     - Besides, these two (thermostats or barostats) are used on **velocity** to guarantee that the simulated trajectories (**coordinates/positions** ) follow the configuration. In this way, we are using NeuralMD to learn the **coordinates/positions** directly. By expectation, we want to claim that such an implicit bias can be learned by our models, especially after we introduce using the Langevin dynamics.
>     - There are also variants of thermostats, e.g., the Nose-Hoover thermostat is deterministic, while the Langevin thermostat is stochastic. Thus, it highly depends on the dataset generation process. To verify it, we have sent emails to the authors of MISATO, and still waiting for their response.
>
>
> |                     | 100-MAE            | 100-MSE            | 100-Stability       | 1k-MAE            | 1k-MSE            | 1k-Stability       | All-MAE            | All-MSE            | All-Stability       |
> |---------------------|----------------|----------------|-----------------|----------------|----------------|-----------------|----------------|----------------|-----------------|
> | VerletMD            | 90.326         | 56.913         | 86.642          | 80.187         | 53.110         | 86.702          | 105.979        | 69.987         | 90.665          |
> | GNN-MD              | 7.176          | 4.726          | 35.431          | 7.787          | 5.118          | 33.926          | 8.260          | 5.456          | 32.638          |
> | DenoisingLD         | 7.112          | 4.684          | 29.956          | 7.746          | 5.090          | 18.898          | 15.878         | 10.544         | 89.586          |
> | NeuralMD-ODE | **6.852** | **4.503** | **19.173** | **7.653** | **5.028** | **15.572** | **8.147** | **5.386** | **17.468** |
> | NeuralMD-SDE  | 6.869          | 4.514          | 19.561          | 7.665          | 5.037          | 16.501          | 8.165          | 5.398          | 19.012          |

---

> ### Author Response · Authors · 2023-11-16
> **Response (2/3)**
>
> | PDB ID | Metric    | VerletMD | GNN-MD    | DenoisingLD | NeuralMD-ODE | NeuralMD-SDE |
> |--------|-----------|----------|-----------|-------------|--------------|--------------|
> | 5WIJ   | MAE       | 9.618    | 2.319     | 2.254       | 2.118        | **2.109**    |
> |        | MSE       | 6.401    | 1.553     | 1.502       | 1.410        | **1.408**    |
> |        | Stability | 79.334   | 45.369    | 18.054      | **12.654**   | 13.340       |
> | 4ZX0   | MAE       | 21.033   | 2.255     | 1.998       | **1.862**    | 1.874        |
> |        | MSE       | 14.109   | 1.520     | 1.347       | **1.260**    | 1.271        |
> |        | Stability | 76.878   | 41.332    | 23.267      | **18.189**   | 18.845       |
> | 3EOV   | MAE       | 25.403   | 3.383     | 3.505       | 3.287        | **3.282**    |
> |        | MSE       | 17.628   | 2.332     | 2.436       | 2.297        | **2.294**    |
> |        | Stability | 91.129   | 57.363    | 51.590      | **44.775**   | 44.800       |
> | 4K6W   | MAE       | 14.682   | 3.674     | 3.555       | 3.503        | **3.429**    |
> |        | MSE       | 9.887    | 2.394     | 2.324       | 2.289        | **2.234**    |
> |        | Stability | 87.147   | 57.852    | 39.580      | 38.562       | **38.476**   |
> | 1KTI   | MAE       | 18.067   | **6.534** | 6.657       | 6.548        | 6.537        |
> |        | MSE       | 12.582   | 4.093     | 4.159       | 4.087        | **4.085**    |
> |        | Stability | 77.315   | 4.691     | 7.377       | 0.525        | **0.463**    |
> | 1XP6   | MAE       | 13.444   | 2.303     | 1.915       | **1.778**    | 1.822        |
> |        | MSE       | 9.559    | 1.505     | 1.282       | **1.182**    | 1.216        |
> |        | Stability | 86.393   | 43.019    | 28.417      | **19.256**   | 22.734       |
> | 4YUR   | MAE       | 15.674   | 7.030     | 6.872       | **6.807**    | 6.826        |
> |        | MSE       | 10.451   | 4.662     | 4.520       | **4.508**    | 4.526        |
> |        | Stability | 81.309   | 50.238    | 32.423      | **23.250**   | 25.008       |
> | 4G3E   | MAE       | 5.181    | 2.672     | 2.577       | 2.548        | **2.478**    |
> |        | MSE       | 3.475    | 1.743     | 1.677       | 1.655        | **1.615**    |
> |        | Stability | 65.377   | 16.365    | 7.188       | **2.113**    | 2.318        |
> | 6B7F   | MAE       | 31.375   | 4.129     | 3.952       | 3.717        | **3.657**    |
> |        | MSE       | 21.920   | 2.759     | 2.676       | 2.503        | **2.469**    |
> |        | Stability | 87.550   | 54.900    | 16.050      | **3.625**    | 22.750       |
> | 3B9S   | MAE       | 19.347   | 2.701     | 2.464       | **2.351**    | 2.374        |
> |        | MSE       | 11.672   | 1.802     | 1.588       | **1.527**    | 1.542        |
> |        | Stability | 41.667   | 43.889    | 8.819       | **0.000**    | **0.000**    |
>
> [1] Allen, Michael P., and Dominic J. Tildesley. Computer simulation of liquids. Oxford university press, 2017.
>
>
> **About performance and baselines**
>
> - First, we want to mention that our model is achieving almost consistently better performance, compared to all the baselines. This is definitely not a perfect solution, but it represents a novel scientific progress in the ML community.
> - Then we want to emphasize that, one insight/goal of this work is to open a novel physics-inspired research line of ML for binding MD prediction, and please allow progressive updates on this.
> - We also have certain questions about RMSF. (1) We do not understand the following sentence: `The RMSF is the best result that can be achieved by a single static structure and is an essential missing baseline.` (2) By RMSF, if you mean `root mean square fluctuation`, then it’s a metric, not a method/baseline. (3) RMSF, as a metric, tells the fluctuation of every single trajectory, thus, it is an unsupervised metric and is not easy to compare with. On the other hand, the stability (proposed by Fu. et al.) metric measures the fluctuation between the predicted trajectory and the ground-truth trajectory, and we want to argue that it is a more comparable metric that includes the fluctuation/stability.
> - Comparison between NeuralMD and DenoisingLD. (1) According to Table 3, NeuralMD is better than DenoisingLD by 10%, 3%, and 70% on three datasets, respectively. We want to argue that this is a consistent and stable improvement. (2) Besides, DenoisingLD is overdamped Langevin dynamics, which is inappropriate for the general MD estimation unless under specific conditions. We leave more details in Sec G in the revised manuscript.

---

> ### Author Response · Authors · 2023-11-16
> **Response (3/3)**
>
> **Changed title**
>
> We are happy to change the title from `group symmetric` to `SE(3)-equivariance`. And we want to point out that the SE(3)-equivariant model for interaction, especially the multi-granularity,  is indeed one core contribution of our work. We spent the whole Sec 3.1 and 3.2 discussing the details.
>
>
> **Answers to the Questions**
> - Clarifications on ML terminologies.
>     - In this work, our usage of the dot product is the same, and you can check [this wiki page](https://en.wikipedia.org/wiki/Dot_product).
>     - We only have scalars and vectors in the input and output, the representations (e.g., $h_i$) are invariant/equivariant representations. We have added the dimension of each variable in Sec E in the revised manuscript.
>     - The number of layers means how many times we repeat the MPNN layer (we omitted the index of layer for simplicity).
>     - The MPNNs are defined in Sec 3.2 explicitly:
>         - The MPNN in BindingNet-Ligand corresponds to Eq 5.
>         - The MPNN in BindingNet-Protein corresponds to the last line of paragraph **Backbone-Level Protein Modeling**.
>         - The MPNN in BindingNet-Complex corresponds to Eq 6.
> - MAE is short for ``mean absolute error’’. We have shown its equation in Sec 3.3 below Eq (8) when it first shows up.
> - When we mean we keep the same backbone, that means we use the BindingNet to provide atom-wise energy/force prediction. This will be fed into the Gaussian distribution in GNN-MD or scoring function in DenoisingLD to learn/estimate $p(x_{t+1}|x_t)$. If readers are interested in these works, we refer to the Related Work paragraph in our manuscript (with references) for an in-depth read. BTW. In ML papers, we often refer the baseline papers to the audience for them to get the details, and leave the main content for our paper’s novel algorithm (Sec 3 in our paper).
> - We had this in the first version. We have mentioned this under **Ablation Study on Flexible Setting** in Sec 4.1, and please check Sec F for more details.
> - The timesteps in the current experiment are fixed (not adaptive) so as to fit the MISATO dataset.
> - Training on the first 80 snapshots, and testing on the last 20 snapshots. The 80th snapshot is given in inference.

---

> ### Comment · Reviewer_oR9H · 2023-11-20
>
> Thanks for the detailed response.
>
> (1) Thanks for clarifying the SE(3) equivariance. However, the Appendix refers to a nearest-neighbor algorithm $x_k$ to construct equivariant frames, deviating from the exposition in the main text. This misalignment should be remedied. I'm also unconvinced by this kind of equivariance accomplished by removing the center of mass as we lose the ability to locally learn equivariant features.
>
> (2) I appreciate the attempt to justify the ODE formulation; however, these are unconvincing as the evaluation metrics are still based on the assumption that one wants to learn an MAE-minimizing simulator. The authors have claimed to capture stochastic dynamics by learning the best deterministic simulator and adding random noise after the fact. This is very different from learning stochastic dynamics.
>
> (3) The RMSF gives an idea of the intrinsic variance of the trajectories and hence the MAE obtained by the best dummy predictor which always predicts a static frame for all timesteps.
>
> Because the problem formulation remains unconvincing and the experimental results remain unimpressive, I will keep the current score.

---

> ### Author Response · Authors · 2023-11-21
> **Response**
>
> Hi reviewer oR9H, thank you for the follow-up. We're pleased to delve deeper into addressing your concerns.
>
> **Q: SE(3)-equivariance**
>
> 1. Thank you for the confirmation. The proof and the main body can match. If you read through Sec C, then you may find that $x_k$ acts as the anchor point. This can hold for the atom nearest to the center of $(x_i, x_j)$, or by the backbone structure of the protein, and the proof still holds. Now in the latest version, we just make $x_i, x_j, x_k$ in Eq 11 exactly the same as the one in Eq 3. The proof of other frames, as highlighted in green in the revised version, is straightforward.
>
> 2. Also, we didn’t claim that we follow a locally equivariant approach for building our method. We are wondering why you insist that we should locally learn equivariant features?  Making our method local is quite straight, we only need to move the local mass center of a subgraph (vectors in each frame) to zero, but the motivation for doing so is unclear to us.
>
>
> **Q: Clarification of algorithm and metric**
>
> Thank you for asking these questions. There are some gaps in the discussion, and we are happy to clarify further.
> - Both the learning and inference of NeuralMD-SDE are stochastic, and it is learning **stochastic dynamics**. We added the random noise and then learned the **stochastic simulator**. The previous revision contained the words 'obtain trajectories' which is confusing, and we have modified it to 'train and sample trajectories`. Please check Sec 3.3 in the latest version, *From Newtonian dynamics to Langevin dynamics*.
> - We are not certain about what this sentence means: `however, these are unconvincing as the evaluation metrics are still based on the assumption that one wants to learn an MAE-minimizing simulator`. Yet, we have tried our best to provide our understanding to you, as follows.
> - All of the works along this research line use MAE as the loss function (related works section), and we added stability metric, following Xiang et al.’s most recent benchmark paper on ML for MD simulation. If you think these are unconvincing, can you provide more insightful explanations on why this is so? This would be appreciated, and then we can know how to solve this issue.
>
>
> **Q: About RMSF**
>
> First thank you for agreeing that RMSF is a metric, not a baseline. In the previous round of discussion, we explained why using RMSF is not ideal as a single-trajectory metric, and why **using stability is a good replacement**. Feel free to check *About performance and baselines* above.

---

### Official Review · Reviewer_ZPQv · 2023-10-28

**Soundness:** 2 fair
**Presentation:** 2 fair
**Contribution:** 2 fair
**Rating:** 5
**Confidence:** 3

**Summary:**

The work proposes a fast numerical MD method for simulating the protein-ligand binding dynamics in a large time interval. This method consists of two main modules: (1) a physics-informed multi-grained group symmetric network to model the protein-ligand complex, and (2) a second-order ODE solver to learn Newtonian mechanics. The proposed method can achieve 2000× speedup compared to the numerical MD methods and outperforms other ML methods on 12 tasks.

**Strengths:**

1.	The author proposed NeuralMD, an ML framework that incorporates a novel multi-grained group symmetric network architecture and second-order ODE Newtonian dynamics, enabling accurate predictions of protein-ligand binding dynamics in a larger time interval.
2.	The authors are the first to explore a large-scale dataset with binding dynamics released in May 2023.
3.	NeuralMD offers a 2000× speedup over standard numerical MD simulation and outperforms other ML approaches by up to ~80% under the stability metric.
4.	NeuralMD not only achieves good performance in single-trajectory binding dynamics predictions, but also has good generalization ability among multiple trajectories.

**Weaknesses:**

1.	This work is based on the first large-scale dataset with binding dynamics, which may have its own limitations.
2.	The paper does not provide a direct comparison with other state-of-the-art methods in terms of computational efficiency.
3.	The article briefly mentions protein-ligand binding dynamics but does not further explain their importance and how such dynamics can be modeled. In addition, the article does not adequately discuss the interactions between proteins and ligands and how these interactions can be incorporated into the simulations.
4.	The paper does not discuss the potential limitations or drawbacks of the NeuralMD framework.
5.	The paper does not delve deeply into the practical implications or real-world applications of the proposed method.
6.	The language of the article is somewhat poorly formulated, with some grammatical errors and spelling mistakes.

**Questions:**

1.	The paper states that the speed of this method is superior to standard numerical MD simulation methods, but this method is only compared with one method and is not compared with ML-based MD simulation methods. Please explain the reasons for this and compare it with more methods to prove the efficiency of the proposed method.
2.	The title of the article mentions a "multi-grained group symmetric framework", but not enough experimental results were provided to prove its effectiveness.
3.	The methods section of the article mentions "BindingNet model that satisfies group symmetry using vector frames and captures the multi-level protein-ligand interactions" but does not explain in detail how this model captures the multi-level interactions. interactions", but does not explain in detail how this model captures the multi-level interactions. Could you provide more details or examples to illustrate this point?
4.	The proposed method only is evaluated on one dataset, which might be specific to the used dataset. The authors need to evaluate their method on other datasets and compare it with more methods.
5.	How does the ML approach discussed in the article compare with other ML methods used for simulating protein-ligand binding dynamics? What is the advantage of the ML method used in this work compared to others?
6.	The article does not describe in detail the specific methods used for the experiments; for example, in the case of protein-ligand binding dynamics simulations, no specific information on the simulation software used, simulation conditions, model parameters, etc. is mentioned. This makes it difficult for the reader to understand and assess the feasibility of the experimental methods.

---

> ### Author Response · Authors · 2023-11-16
> **Response (1/3)**
>
> Thank you for acknowledging our work as the first to explore large-scale binding MD datasets, solid performance, and efficiency. We believe that we have attempted to solve your concerns during the rebuttal and in the revised manuscript. We have listed the details below.
>
> **Dataset limitations**
>
> The binding dynamics dataset issue is out of the scope of our work. We also want to mention that this is actually the most critical bottleneck that prevents ML researchers from developing tools for domain-specific tasks. If there exist any other public binding MD datasets, we are happy to use them.
>
> **Efficiency with other DL models**
>
> Thank you for raising this. We have now added them below (FPS, corresponding to Table ). The main bottleneck of using ML for binding MD simulation is on the backbone model, and since we are using the same backbone model in our experiments, i.e., the running time of these DL models is on the **same order of magnitude**. That’s why we didn’t highlight the time comparison among different DL models in the first version.
>
> | PDB ID          | 5WIJ   | 4ZX0   | 3EOV   | 4K6W   | 1KTI   | 1XP6   | 4YUR   | 4G3E   | 6B7F   | 3B9S   | Average |
> |-----------------|--------|--------|--------|--------|--------|--------|--------|--------|--------|--------|---------|
> | VerletMD        | 12.564 | 30.320 | 29.890 | 26.011 | 19.812 | 28.023 | 31.513 | 29.557 | 19.442 | 31.182 | 25.831  |
> | GNN MD          | 22.639 | 41.083 | 26.219 | 26.798 | 22.435 | 35.406 | 25.370 | 36.677 | 42.018 | 40.339 | 31.898  |
> | Denoising LD    | 37.047 | 39.764 | 38.232 | 32.222 | 22.181 | 34.501 | 41.498 | 38.003 | 42.261 | 41.867 | 36.758  |
> | NeuralMD (Ours) | 33.164 | 39.415 | 31.720 | 31.909 | 24.566 | 37.135 | 39.365 | 39.172 | 20.320 | 37.202 | 33.397  |
>
>
> **How to model the protein-ligand interactions**
>
> This is an important module, and we have shown it in Sec 3.1 and 3.2 in the first version. To be more specific:
> - In Eq 4 (Sec 3.1), we illustrated how the vector frame ($F_{complex}$) is constructed for protein-ligand interactions.
> - In Eq 6 (Sec 3.2), we showed the MPNN module for protein-ligand binding, based on $F_{complex}$.
> - We also showed the pipeline in Figure 4 & 5 (appendix).
> - We discussed how the interacted forces can be used for the binding MD estimation in Eq 6 and the following paragraph (Sec 3.2). So the output is atom-level forces, which will be fed into the Newtonian dynamics (Eq 8) for trajectory estimation.
> - We added the model architecture in more detail in Sec E in the revised version.

---

> ### Author Response · Authors · 2023-11-16
> **Response (2/3)**
>
> **Potential limitation**
> One potential limitation, which has been raised by reviewer oR9H is that we didn’t consider the solvent molecules in the MD system. This is because the public dataset does not include this information. Meanwhile, we are still able to handle such a limitation by adding random terms into the predicted forces, i.e., the Langevin dynamics. We have added the results of *single trajectories* below. Further in the revised manuscript, we added one paragraph in Sec 3.3 for a detailed explanation and the results of *multiple trajectories* in Sec 4. We can observe that the performance of ODE and SDE are very close to each other.
>
> | PDB ID | Metric    | VerletMD | GNN-MD    | DenoisingLD | NeuralMD-ODE | NeuralMD-SDE |
> |--------|-----------|----------|-----------|-------------|--------------|--------------|
> | 5WIJ   | MAE       | 9.618    | 2.319     | 2.254       | 2.118        | **2.109**    |
> |        | MSE       | 6.401    | 1.553     | 1.502       | 1.410        | **1.408**    |
> |        | Stability | 79.334   | 45.369    | 18.054      | **12.654**   | 13.340       |
> | 4ZX0   | MAE       | 21.033   | 2.255     | 1.998       | **1.862**    | 1.874        |
> |        | MSE       | 14.109   | 1.520     | 1.347       | **1.260**    | 1.271        |
> |        | Stability | 76.878   | 41.332    | 23.267      | **18.189**   | 18.845       |
> | 3EOV   | MAE       | 25.403   | 3.383     | 3.505       | 3.287        | **3.282**    |
> |        | MSE       | 17.628   | 2.332     | 2.436       | 2.297        | **2.294**    |
> |        | Stability | 91.129   | 57.363    | 51.590      | **44.775**   | 44.800       |
> | 4K6W   | MAE       | 14.682   | 3.674     | 3.555       | 3.503        | **3.429**    |
> |        | MSE       | 9.887    | 2.394     | 2.324       | 2.289        | **2.234**    |
> |        | Stability | 87.147   | 57.852    | 39.580      | 38.562       | **38.476**   |
> | 1KTI   | MAE       | 18.067   | **6.534** | 6.657       | 6.548        | 6.537        |
> |        | MSE       | 12.582   | 4.093     | 4.159       | 4.087        | **4.085**    |
> |        | Stability | 77.315   | 4.691     | 7.377       | 0.525        | **0.463**    |
> | 1XP6   | MAE       | 13.444   | 2.303     | 1.915       | **1.778**    | 1.822        |
> |        | MSE       | 9.559    | 1.505     | 1.282       | **1.182**    | 1.216        |
> |        | Stability | 86.393   | 43.019    | 28.417      | **19.256**   | 22.734       |
> | 4YUR   | MAE       | 15.674   | 7.030     | 6.872       | **6.807**    | 6.826        |
> |        | MSE       | 10.451   | 4.662     | 4.520       | **4.508**    | 4.526        |
> |        | Stability | 81.309   | 50.238    | 32.423      | **23.250**   | 25.008       |
> | 4G3E   | MAE       | 5.181    | 2.672     | 2.577       | 2.548        | **2.478**    |
> |        | MSE       | 3.475    | 1.743     | 1.677       | 1.655        | **1.615**    |
> |        | Stability | 65.377   | 16.365    | 7.188       | **2.113**    | 2.318        |
> | 6B7F   | MAE       | 31.375   | 4.129     | 3.952       | 3.717        | **3.657**    |
> |        | MSE       | 21.920   | 2.759     | 2.676       | 2.503        | **2.469**    |
> |        | Stability | 87.550   | 54.900    | 16.050      | **3.625**    | 22.750       |
> | 3B9S   | MAE       | 19.347   | 2.701     | 2.464       | **2.351**    | 2.374        |
> |        | MSE       | 11.672   | 1.802     | 1.588       | **1.527**    | 1.542        |
> |        | Stability | 41.667   | 43.889    | 8.819       | **0.000**    | **0.000**    |
>
>
>
> **Practical implications**
>
> The experiments of binding MD estimation on MISATO (a subset of PDB) have practical implications. If we want to extend this to other real-world applications, that means we need other binding MD datasets, and we are happy to check if the reviewer can provide specific references.
>
> **Effectiveness of multi-grained**
>
> The proof of this is trivial because, without the multi-grained modeling, the model cannot fit into the GPU memory.
>
> **Multi-level interactions**
>
> So we have the multi-grained modeling discussed in Sec 3.1 & 3.2.
> - In Sec 3.1, we have shown three levels of vector frames in Eq 2, 3, 4.
> - In Sec 3.2, we have shown how to do MPNN on these three vector frames, respectively.
> - Finally, the atom forces are used to get the trajectory following Newtonian dynamics, $a = F/m$.
> - This is also illustrated in Figure 4.
> - We added more details in Sec E of the revised manuscript.

---

> ### Author Response · Authors · 2023-11-16
> **Response (3/3)**
>
> **More binding datasets and more methods**
>
> - Thank you for raising this question. First, we want to highlight that we are not just using one dataset. As also confirmed by your comments, we include 10 single binding trajectories and 3 datasets on multiple trajectories. These trajectories are from PDB, the most comprehensive dataset for protein (binding), thus, we aim to assert that the experiments conducted in this study are indicative and representative of a substantial volume of binding MD data.
> - Then as also confirmed by your comments, the dataset is one limitation in AI for binding MD estimation, and MISATO is the first large-scale binding MD dataset released in May 2023. If you can provide other existing public large-scale binding MD datasets, we are happy to try them in future work.
> - For baselines, if the reviewer can list the specific papers, we are happy to compare.
>
> **Model difference with existing works**
>
> The other models for binding MD trajectory prediction can be categorized into two venues:
> - No incorporation of physics rules, like GNN-MD.
> - Inappropriate incorporation of physical rules, like overdamped Langevine dynamics in the DenoisingLD papers. We provide more details in Sec G of the revised manuscript. Feel free to check.
> - Thus, in comparison, we can better incorporate the physical rules (Newtonian dynamics and the newly added Langevin dynamics), and such inductive bias can help better model the binding MD.
>
> **Dataset simulation illustration is out of the scope of this work**
>
> We confirmed that MISATO uses the Amber20 software suite and NVT, and we provide more dataset statistics (number of atoms and number of residues, etc.) in Sec D. However, we want to highlight that doing simulation is beyond the scope of our work, because we are not the authors of MISATO. The credit for the binding MD dataset generation goes to the MISATO paper.

---

### Official Review · Reviewer_582T · 2023-10-31

**Soundness:** 3 good
**Presentation:** 3 good
**Contribution:** 3 good
**Rating:** 6
**Confidence:** 3

**Summary:**

This paper proposes a geometric deep learning framework for MD simulation of protein-ligand complexes. It is composed of a BindingNet model to represent a protein-ligand complex at multiple levels and a neural ordinary differential equation (ODE) solver to predict the trajectory under Newtonian mechanics. The method is evaluated on a recent large-scale MD simulation benchmark MISATO and shows state-of-the-art performance on multiple benchmarks. Importantly, it achieves a 2000x speedup over standard numerical MD simulation methods.

**Strengths:**

* This paper proposes a physics-informed architecture that involves Newtonian mechanics in trajectory inference.
* This paper presents a comprehensive evaluation on protein-ligand binding MD simulation benchmarks.
* The proposed model shows state-of-the-art performance on multiple benchmarks.
* The proposed method is scalable, with 2000x speed up compared to standard MD simulation tools.

**Weaknesses:**

* This paper lacks ablation studies illustrating the benefit of different components.
* The description of evaluation metric is not crystal clear (see questions below)

**Questions:**

1. Can you present ablation studies by replacing your BindingNet architecture with other existing geometric architectures such as EquiFormer and EGNN?
2. Can you conduct ablation studies to understand the benefit of multi-level protein-ligand representation? I am not sure if having multi-level representation is helpful
3. For the MAE and MSE metric, is the MAE/MSE calculated over the whole trajectory (every time step) or only the final step?

---

> ### Author Response · Authors · 2023-11-16
> **Response**
>
> We appreciate your recognition of our work on physics-inspired architecture, comprehensive evaluation, and efficiency. Your primary concerns are about backbone models and ablation studies. We believe that the following explanations adequately address the points of concern you raised.
>
> **Ablation studies on different components**
>
> Thank you for raising this question. We assume that your question is about if we can test the three components of BindingNet separately, as shown in Figure 4. The answer is no, because separating them would destroy the physical rules in MD. Let us explain in more detail:
> - The BindingNet-Ligand models the interactions within the ligand itself.
> - The BindingNet-Protein and BindingNet-Complex learn the atom-level forces induced by the proteins.
> - By combining these modules, we can learn the atom forces from all the other atoms, which is the physical-inspired MD estimation.
> - On the other hand, we also added an ablation study on introducing stochastic terms induced by the solvent molecules (they were considered in the dataset generation, but not provided to the public). We have attached the results in the revised manuscript. Please feel free to check it.
>
> **Can we replace BindingNet with Equiformer and EGNN, and why multi-level?**
>
> These two questions are highly connected, so we answer them together. In practice, the multi-grained modeling is appealing from the engineering perspective. If we model the large molecular systems, even for one ligand-protein complex, there are on average 1k residues (around 20k-30k atoms) in the MISATO dataset. Thus, modeling such a big molecular system is challenging for GPU memory. This means:
> - We cannot replace BindingNet with Equiformer or EGNN.
> - The multi-grained (or multi-level) modeling is motivated by reducing the computational cost, and without it, the GPU memory cannot hold for such a large molecular system.
> - Actually, even in our current system, we are taking batch size with 2 so as to fit the GPU memory.
>
> **Clarification on evaluation**
>
> Yes, both the MAE and MSE are calculated over the whole trajectory snapshots in the test set, not only the final snapshot. We have this highlighted in Sec 4.

---

> ### Comment · Reviewer_582T · 2023-11-21
> **Thank you for your response**
>
> I would like to thank the reviewers for their response. I have two comments:
>
> 1. I would probably disagree that one cannot replace BindingNet with EGNN because we have been training EGNN internally on the MISATO dataset and it runs pretty fast. Due to the lack of baselines, it is a bit unclear whether BindingNet is superior than existing geometric encoders.
>
> 2. I initially thought that MAE/MSE was calculated at the last snapshot. If it was calculated over the whole trajectory (every snapshot), I would imagine the MAE/MSE to be substantially lower (currently MAE is around 8.1 for MISATO-all) because a ligand wouldn't move that much during MD simulation. What's the MAE if you just use the ligand pose at time 0? (meaning, the ligand doesn't move at all during MD simulation). This is probably related to RMSF metric the other reviewer brought up

---

> ### Author Response · Authors · 2023-11-21
> **Reply**
>
> Hi reviewer 582T,
>
> Thank you for the follow-up. We are happy to address your questions in more detail.
>
> 1. We list the main reasons why EGNN is not a good fit below.
>     - [Memory] We agree that EGNN is efficient, but the bottleneck here is the computational memory. (The original EGNN is connecting all atoms)
>     - [Math correctness] EGNN is E(3)-equivariant. However, when the protein is fixed, the modeling of the ligand needs to be SE(3)-equivariant and reflection-antisymmetric. We explained more details in Sec C.
>     - [Performance] Further on the performance, the Geom3D benchmark has provided around 50 geometric tasks, showing how the SE(3)-equivariant models outperform EGNN / GVP.
>
> 2. Thank you for raising this question, and let's explain more insights to you.
>     - If we focus on the **single-trajectory** setting (Table 1), then you can see that the MAE and MSE are comparatively small. This is also the setting that most baseline papers are working on.
>     - In this paper, we introduce another challenging task of generalization among **multiple trajectories** (Table 2), this is where you see that the MAE & MSE are much larger. Notice that here, the MAE & MSE are the averages of all the snapshots and all the testing trajectories.
>     - Thank you for raising this question on **RMSF**. RMSF is the metric of a single trajectory, and if we want to compare the fluctuation between the predicted trajectories and ground-truth trajectories, then that is the **stability** metric. In other words, we have already included the RMSF on prediction-and-true trajectories.
>
> Hope this answers your questions, and we are happy to provide more insights.

---

> > ### Comment · Reviewer_582T · 2023-11-21
> > **Reply**
> >
> > Thank you for your reply. I agree with you on E(3) vs SE(3) equivariance. It would be nice to compare with other SE(3)-equivariant models, nevertheless.
> >
> > I totally understand that multi-trajectory evaluation is much more challenging (imo, this is much more useful in practice). Still, what's the performance if your model just do nothing and output the ligand pose at time 0?

---

> ### Author Response · Authors · 2023-11-22
> **Reply**
>
> Hi Reviewer 582T,
>
> Thank you for the follow-up!
>
> 1. Yes, we are definitely going to explore more SE(3)-equivariant models in the future. BTW. We also share some insights that another motivation for using vector frame modeling is for certain tasks (e.g., protein representation), the vector frame modeling outperforms other methods by a large margin. You can check this [CDConv paper](https://arxiv.org/abs/2210.08511) in case interested.
>
> 2. Thank you for the suggestion. So we called this baseline `Void MD` since we are just taking the first frame as the prediction for the rest frames for each trajectory. We got some interesting results, as shown below. As you can see, our NeuralMD is still reaching the best performance, while all the existing/published baselines are worse than Void MD. We also acknowledge the marginal performance improvement, but we want to point out the challenge of this task, and we reach a **consistent** improvement. This is a promising sign to prove the better generalization ability of NeuralMD, especially compared to the published baselines.
>
> | | GNN MD| DenoisingLD | Void MD | NeuralMD (Ours) |
> | :--:| :--:| :--:| :--:| :--:|
> | MISATO-100 | 7.176 | 7.112 | 7.048 | **6.852** |
> | MISATO-1000 | 7.787 | 7.746 | 7.723 | **7.653** |
> | MISATO-All | 8.260 | 15.878 | 8.206 | **8.147** |
>
>
> We hope this answers your questions, and any re-evaluation is appreciated.
>
> Regards,
>
> Authors of NeuralMD

---

### Official Review · Reviewer_K9WG · 2023-11-04

**Soundness:** 3 good
**Presentation:** 3 good
**Contribution:** 3 good
**Rating:** 6
**Confidence:** 5

**Summary:**

Authors  propose a principled approach that incorporates a novel physics-informed multi-grained group symmetric framework. Specifically, we propose (1) a BindingNet model that satisfies group symmetry using vector frames and captures the multi-level protein-ligand interactions, and (2) an augmented neural ordinary differential equation solver that learns the trajectory under Newtonian mechanics.

Authors devised NeuralMD, an ML framework that incorporates a novel multi-grained group symmetric network architecture and second-order ODE Newtonian dynamics, enabling accurate predictions of protein-ligand binding dynamics in a larger time interval.

**Strengths:**

1. Authors have quantitatively and qualitatively verifed that NeuralMD achieves superior performance on 13 binding prediction tasks.
2. Authors showed the efficiency and effectiveness of NeuralMD, with a 2000× speedup over standard numerical MD simulation and outperforming all other ML approaches by up to ~80% under the stability metric.

**Weaknesses:**

One potential limitation of this work is the dataset. Currently, authors are using the MISATO dataset, a binding simulation dataset with a large timescale. However, NeuralMD is agnostic to the time interval, and it can also be applied to binding dynamics datasets with time interval as a femtosecond.

**Questions:**

1. Authors qualitatively show that NeuralMD reaches more stable binding predictions. Is there a way to shoe quantitatively as well.
2.

---

> ### Author Response · Authors · 2023-11-16
> **Response**
>
> We appreciate your positive comments on our work. Your primary concerns are about the dataset and evaluation, and we have explained them below.
>
> 1. **[regarding the dataset limitation]** We agree that the dataset is one of the potential limitations here; as highlighted in the conclusion section, making more datasets available may need help and support from the whole community. Creating such a dataset not only is very costly, but also requires sufficient domain expertise from different domains, which makes it challenging. As stated in the paper, MISATO dataset is the only publicly available large-scale dataset, and that is why we empirically showed the potential of our NeuralMD method using this dataset. Regarding its effect on femtosecond MD, we unfortunately can’t perform any empirical studies at this stage, although we theoretically show that our method is agnostic to the MD time interval.
>
> 2. **[regarding quantitative evaluation]** Yes, we followed the *stability metric* from previous work, and reported those quantitative results in Tables 1 & 3.

---

### Author Response · Authors · 2023-11-19
**General Reply and Gentle Reminder**

Hi Reviewers K9WG, 582T, ZPQv, oR9H,

Hope this message finds you well. We have carefully addressed your questions and have revised the manuscript based on your valuable advice:
- [NeuralMD SDE] Added the random noise term to mimic the missing solvent molecules and Langevin thermostats (confirmed with the authors).
- [Experiments] Ran the additional experiments of NeuralMD SDE, and the updated results are in Tables 1 & 3 in the revised manuscript.
- [Architecture clarification] Explained the multi-grained architecture of BindingNet, especially the details in Sec E.
- [Math clarification] Explained the math is correct: SE(3)-equivariance is guaranteed by removing the mass center.
- [Running time] Added running time of all the baselines.

We are eager to know if our response has effectively addressed the concerns you raised in your initial reviews. Should you need further clarification or have any additional points you would like us to consider, please do not hesitate to share your thoughts. We are committed to ensuring that all your concerns are fully addressed.

Regards,

Authors of NeuralMD

---

> ### Author Response · Authors · 2023-11-22
> **Gentle Reminder**
>
> Hi Reviewers K9WG, 582T, ZPQv, oR9H,
>
> Since the deadline is approaching today, should there be any remaining unclear statements, please don't hesitate to inform us. We'll make every effort to address your concerns.
>
> Regards,
>
> Authors of NeuralMD

---

### Author Response · Authors · 2023-11-23
**Summary of Rebuttal**

Hi reviewers and ACs,

We fully recognize the demanding schedules of everyone involved. Therefore, we have compiled the main concerns raised by the reviewers along with our respective resolutions, should they be beneficial for your upcoming discussions or next steps.

1. **[Math]** We clarified the SE(3)-equivariance proof, especially in the appendix, which can exactly match one of the vector frames in the main body. The other two frames' equivariance property can be derived similarly.
2. **[ODE & SDE]** We have added both Newtonian dynamics (ODE) and Langevin dynamics (SDE) experiments. Note that for Langevin dynamics, we added the noise terms for both training and inference, which means it is stochastic dynamics.
3. **[Evaluation and Void MD baseline]** The `RMSF(traj)` cannot be used as a metric, and we use `stability(pred traj, true traj)` as a surrogate on fluctuation. Further, as suggested by reviewer 582T, we added a new baseline (Void MD) where each predicted snapshot is the first snapshot, and the results are shown below:
| | GNN MD| DenoisingLD | Void MD | NeuralMD (Ours) |
| :--:| :--:| :--:| :--:| :--:|
| MISATO-100 | 7.176 | 7.112 | 7.048 | **6.852** |
| MISATO-1000 | 7.787 | 7.746 | 7.723 | **7.653** |
| MISATO-All | 8.260 | 15.878 | 8.206 | **8.147** |
4. **[Performance and Contribution]** We agree that NeuralMD is not a perfect model, but it reaches consistent improvements on almost all the single- and multi-trajectory prediction tasks. Such consistent enhancement stands as a promising indication of the model's generalization capabilities of NeuralMD. Particularly in contexts where prevailing works might overlook physical mechanics (e.g., overdamped Langevin dynamics in DDPM does not fit MD), we insist on the novelty and contribution of NeuralMD.
5. **[Datasets]** To the best of our knowledge, the MISATO dataset is the largest binding dynamics dataset. While we remain open to testing additional datasets in the future, expanding beyond the 13 binding tasks covered in this current paper falls outside the scope of this study.

Thank you again for your comments and thoughtful reconsideration.

Regards,

Authors of NeuralMD

---

### Meta-Review · Area_Chair_xxP3 · 2023-12-15

**Metareview:**

The paper tries to learn and predict the molecular dynamics of the protein ligand bindings. It uses three separate equivariant representations to represent protein, ligand and their interactions at different scales – atomistic and residue level. Then there are separate networks operating on each of those three representations that, given the current state, predicts the force to model Newtonian dynamics. All the predicted forces are given to a neural ODE to solve for the updated state. The method is benchmarked for both single and multi trajectories with metrics such as MAE, MSE, and stability, on tasks including the recently-released MISATO dataset.


The paper has been thoroughly reviewed and discussed with the authors from various aspects. As a result, during the discussion period, several clarifications and new results have been added to the revised paper, importantly including an SDE-based model for trajectories that are perturbed with stochastic noise and a key baseline with static initial state.


On the positive side, the AC believes the paper has clear merits as it is novel and plausible in approach and progressive in the set of experiments. However, as initially was pointed out by reviewers and discussed with later on, more empirical side studies including metrics and baselines is required to clearly show when and how it works.


For a next version, the AC suggests that the authors add more empirical analysis, including using the RMSF metric, and analyze the obtained quantities by NeuralMD in comparison with other methods but more importantly self-constructed informative baselines such as the static predictor that was added during the discussion period. More can be done there, for instance by replacing the initial state with one predicted marginal state that might improve the baseline. Shying away from providing established metrics and baselines only raises doubts that undermines the efforts of the work. The discussions, for instance on the downside of metrics, can be reserved for analyzing the obtained results and conditioned on the observed quantities which will be more conducive to substantiated conclusions. The task is extremely hard and so consistent improvement on every possible metric is unreasonable to expect but shedding full light on when and how the method works is a requirement for conclusivity and further development.


The AC, therefore, recommends rejection for this submission but believes the paper can become an excellent candidate for a next venue if it covers more aspects in its empirical analysis.

**Justification For Why Not Higher Score:**

Despite all the positive aspects outlined in the meta review, the work abstains from reporting on certain aspects of the proposed method which is important for the full picture of the empirical aspect for an application paper. Therefore, it has to be improved before it can be accepted.

**Justification For Why Not Lower Score:**

N/A

---

### Decision · Program_Chairs · 2024-01-16

Reject